# Arabidopsis CaLB1 undergoes phase separation with the ESCRT protein ALIX and modulates autophagosome maturation

Niccolò Mosesso [1], Niharika Savant Lerner[1,2], Tobias Bläske[1], Felix Groh[1], Shane Maguire [2,3], Marie Laura Niedermeier[2,4], Eliane Landwehr [2,5], Karin Vogel [1], Konstanze Meergans[1], Marie-Kristin Nagel[1], Malte Drescher[2,5], Florian Stengel [2,4], Karin Hauser [2,3] & Erika Isono [1,2,6] ✉

Autophagy is relevant for diverse processes in eukaryotic cells, making its regulation of fundamental importance. The formation and maturation of autophagosomes require a complex choreography of numerous factors. The endosomal sorting complex required for transport (ESCRT) is implicated in the final step of autophagosomal maturation by sealing of the phagophore membrane. ESCRT-III components were shown to mediate membrane scission by forming filaments that interact with cellular membranes. However, the molecular mechanisms underlying the recruitment of ESCRTs to non-endosomal membranes remain largely unknown. Here we focus on the ESCRT-associated protein ALG2-interacting protein X (ALIX) and identify $Ca^{2+}$-dependent lipid binding protein 1 (CaLB1) as its interactor. Our findings demonstrate that CaLB1 interacts with AUTOPHAGY8 (ATG8) and PI(3)P, a phospholipid found in autophagosomal membranes. Moreover, CaLB1 and ALIX localize with ATG8 on autophagosomes upon salt treatment and assemble together into condensates. The depletion of CaLB1 impacts the maturation of salt-induced autophagosomes and leads to reduced delivery of autophagosomes to the vacuole. Here, we propose a crucial role of CaLB1 in augmenting phase separation of ALIX, facilitating the recruitment of ESCRT-III to the site of phagophore closure thereby ensuring efficient maturation of autophagosomes.

Adaptation to changing local and global environmental conditions is imperative for all organisms, particularly for sessile organisms like plants. Adverse conditions, such as salt stress, lead to the accumulation of damaged proteins, protein aggregates, and damaged organelles in the cell, which can cause proteotoxicity or impair cellular homeostasis. To survive salt stress, it is thus essential to eliminate these non-functional molecular aggregates, which is often mediated by selective degradation pathways including autophagy. Thus, its regulation is of fundamental importance for higher eukaryotes[1].

AUTOPHAGY-RELATED PROTEINS (ATGs) ensure the maturation of a growing phagophore, a cup-shaped membrane that engulfs unfunctional organelles, protein aggregates, and other

[1]Plant Physiology and Biochemistry, Department of Biology, University of Konstanz, Universitätsstraße 10, 78457 Konstanz, Germany. [2]Konstanz Research School Chemical Biology, University of Konstanz, Universitätsstraße 10, 78457 Konstanz, Germany. [3]Biophysical Chemistry, Department of Chemistry, University of Konstanz, Universitätsstraße 10, 78457 Konstanz, Germany. [4]Biochemistry and Mass Spectrometry, Department of Biology, University of Konstanz, Universitätsstraße 10, 78457 Konstanz, Germany. [5]Spectroscopy of Complex Systems, Department of Chemistry, University of Konstanz, Universitätsstraße 10, 78457 Konstanz, Germany. [6]Division of Molecular Cell Biology, National Institute for Basic Biology, Nishigonaka 38, Myodaiji, Okazaki 444-8585 Aichi, Japan. ✉e-mail: erika.isono@uni-konstanz.de

macromolecules for their subsequent transport to the vacuole for degradation[2]. Among the various ATG proteins, ATG8 populates the phagophore, coordinating the formation of autophagosomes and acting as a hub for the recruitment of receptors and adaptors which are essential for the selective recognition of autophagic cargos and help mediate the trafficking and fusion of the autophagosomes, respectively. In the final stage of autophagosome maturation, the phagophore membrane must be sealed, a process in which the endosomal sorting complex required for transport (ESCRT) machinery is implicated[3–7].

Among the four ESCRT complexes, ESCRT-0, ESCRT-I, ESCRT-II, and ESCRT III, subunits of ESCRT-III were shown to form filaments that could arbitrate membrane repair, scission and closure[8–10]. The accurate spatio-temporal recruitment of ESCRT-III is thus key to many cellular and physiological processes that require membrane remodeling. ESCRT-III recruitment has been extensively studied at the endosomal membrane, where ESCRT-III is positioned by ESCRT-I and ESCRT-II or other ESCRT-associated proteins[9,11]. Recent studies have shown that ESCRT-III can be recruited to the nuclear membrane through interactions with proteins residing at the inner nuclear membrane[9]. However, our current understanding still lacks molecular knowledge as to how ESCRTs are recruited to other membrane compartments including the autophagosome.

ALIX is a conserved ESCRT-associated protein[12]. Arabidopsis ALIX, the yeast ortholog BCK1-like resistance to osmotic shock 1p (Bro1p), and mammalian ALIX, all have three conserved domains: A V-shaped domain (V-domain) required for sorting of ubiquitylated endosomal cargos by binding to ubiquitin, a proline-rich domain (PRD) involved in protein-protein interactions and ALIX activation, and a Bro1 domain essential for the interaction with ESCRT-III. By binding ESCRT-III, ALIX serves in diverse cellular processes that require ESCRT activity. Arabidopsis ALIX is implicated in endosomal cargo sorting as well as sorting of vacuolar proteins[13–15]. Human ALIX was shown to interact with the autophagic ATG12-ATG3 complex and is required for bulk autophagy[16]. However, the underlying mechanisms of ALIX recruitment to the autophagosomes are not well understood.

In this work, we identify Ca$^{2+}$-dependent lipid binding protein 1 (CaLB1) as an interactor of ALIX. CaLB1 interacts with the phosphoinositide PI(3)P and the ubiquitin-like decorator of autophagic membrane ATG8, colocalizes on autophagosomes together with ALIX, and is required for autophagic turnover during salt-induced autophagy. CaLB1 and ALIX undergo phase separation and assemble in molecular condensates, while the intrinsically disordered region of CaLB1 stimulates the condensation of ALIX. CaLB1 can thus be important for the proper positioning of ESCRT on autophagosomes, thereby ensuring the efficient autophagosome maturation and delivery of autophagic cargo to the vacuole.

## Results

### ALIX interacts with CaLB1
We first wanted to establish whether Arabidopsis ALIX is required for autophagy and investigated the accumulation of ATG8 and NEIGHBOR OF BRCA1 GENE1 (NBR1) in wild type and *alix* null mutants (Fig. 1a). ATG8 is a ubiquitin-like protein that decorates the autophagic membranes and NBR1 is an autophagic receptor that binds to ATG8[17]. Both are transported to and degraded in the vacuole together with autophagic cargos. We investigated the level of ATG8 and NBR1 and found that both ATG8 and NBR1 accumulated in *alix-2* and *alix-4*, as observed in the autophagy-defective *atg7-2* (Fig. 1a), suggesting that ALIX is necessary for proper autophagic turnover.

Next, to investigate the molecular function of ALIX in autophagy, we conducted a yeast two-hybrid screening using ALIX as bait against an Arabidopsis ORF library[18] and identified a Ca$^{2+}$-dependent lipid binding (CaLB) protein, which we designated as CaLB1 (AT4G34150), as an interactor of ALIX. CaLB1 has an amino (N)-terminal type-II C2

domain and a carboxy (C)-terminal proline-rich domain (PRD) (Fig. 1b). The closest homologs, AT1G63220 and AT3G55470, share 24.3% and 23.6% amino acid identity with CaLB1, respectively, and lack a PRD (Supplementary Fig. 1a, b). Sequence homologs of CaLB1 can be found in other plant species but not in opisthokonts (Supplementary Fig. 1c).

To identify the region responsible for the interaction between ALIX and CaLB1, we performed a YTH analysis using fragments of Arabidopsis ALIX and CaLB1 (Fig. 1b–e and Supplementary Fig. 1d, e). In an in vitro pull-down assay, CaLB1-6xHis interacted with the MBP-tagged ALIX(ΔBRO1) lacking the BRO1 domain, however, not with MBP-ALIX (BRO1) lacking the PRD (Fig. 1f and Supplementary Fig. 1f). Similarly, GST-tagged ALIX interacted with full-length CaLB1 but not with CaLB1(C2) lacking the PRD (Fig. 1g and Supplementary Fig. 1g), showing that CaLB1 and ALIX interact through their PRDs.

For investigating the colocalization of CaLB1 and ALIX *in planta*, we generated plants expressing *CaLB1pro: CaLB1-mRFP* together with *ALIXpro: GFP-ALIX*. Confocal microscopy showed that CaLB1-mRFP is localized to the cytosol and cytosolic foci as ALIX[14]. 35.3% of CaLB1-mRFP foci colocalized with GFP-ALIX (Fig. 1h), indicating that CaLB1 and ALIX can function in the same cellular compartment.

### CaLB1 and ALIX bind to PI(3)P
The C2 domain of CaLB1 has three putative Ca$^{2+}$-binding regions (CBRs) (Fig. 2a). The C2 domain was first characterized in protein kinase C and is a globular membrane-binding domain conserved widely in eukaryotes. The majority of C2 domain-containing proteins bind membrane lipids in a Ca$^{2+}$-dependent manner and have versatile functions in signal transduction, membrane trafficking, and membrane fusion[19,20].

Microscale thermophoresis (MST) analyses showed that CaLB1 (C2) binds Ca$^{2+}$ but not Mg$^{2+}$ (Fig. 2b and Supplementary Fig. 2a). As the full-length CaLB1 precipitated at the concentration required for the MST analysis, the binding studies could be conducted only with the C2 domain. When an aspartate residue in one of the predicted Ca$^{2+}$-binding loop was mutated, the resulting CaLB1[C2(D32A)] variant showed reduced affinity for Ca$^{2+}$ (Fig. 2b and Supplementary Fig. 2a).

To analyze whether binding of Ca$^{2+}$ influences the conformation of CaLB1, three recombinant CaLB1(C2) variants bearing the native cysteine C19 and a second labeling site S29C, N55C, or L83C were spin-labeled and analyzed by double electron-electron resonance (DEER) spectroscopy[21] (Supplementary Fig. 2b). Changes in the overall protein structure were not observed; however, a narrowing of the distance observed for S29C and N55C variants in the presence of Ca$^{2+}$ suggests a possible Ca$^{2+}$-mediated structural stabilization of the C2 domain (Supplementary Fig. 2c–e).

As C2 domains are known membrane-binding domains, we wanted to know whether CaLB1 can also bind membrane lipids and performed a lipid protein overlay assay using recombinant CaLB1-6xHis in the presence of either 10 μM Ca$^{2+}$ or 5 mM EGTA. Among the tested phospholipids, CaLB1 bound preferentially to phosphatidylinositol-3-phosphate [PI(3)P] in a Ca$^{2+}$-dependent manner (Fig. 2c and Supplementary Fig. 2f).

CaLB1 can interact with phosphoinositides integrated into membrane bilayers in a Ca$^{2+}$ concentration-dependent manner[22]. To understand whether CaLB1 membrane binding is dependent on PI(3)P, we performed attenuated total reflection-Fourier transform infrared spectroscopy (ATR-FTIR) spectroscopy[23,24] on a solid supported lipid bilayer (SSLB) containing either PI(3)P or phosphatidylinositol-4-phosphate [PI(4)P] in the presence of 20 μM Ca$^{2+}$. The affinity to the membrane was reduced when the Ca$^{2+}$-binding loop mutant variant CaLB1(D32A) was used instead of wild-type CaLB1, or when PI(4)P-containing membranes were used (Fig. 2d and Supplementary Fig. 2g), indicating that CaLB1 binds preferentially to PI(3)P-containing membranes and that Ca$^{2+}$ is important for this interaction.

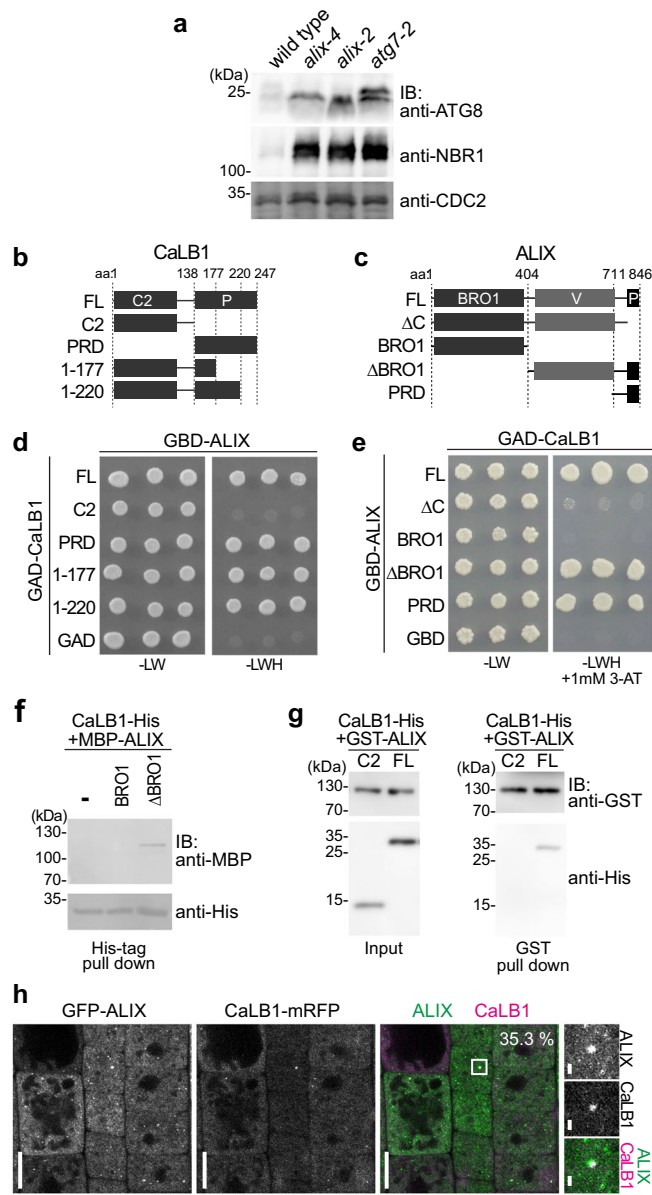

**Fig. 1 | CaLB1 interacts with ALIX. a** *alix* mutants accumulate autophagosome markers. Total extracts of 7-day-old wild type, *alix-4, alix-2* and *atg7-2* were subjected to immunoblotting using an anti-ATG8 and anti-NBR1 antibodies. An anti-CDC2-antibody was used as a loading control. The experiment was conducted twice and one of the results is shown. Constructs of CaLB1 (**b**) and Arabidopsis ALIX (**c**) used for the interaction studies. FL: full length; C2: C2 domain; P/PRD: proline-rich domain; 1-177: CaLB1 fragment 1-177; 1-220: CaLB1 fragment 1-220; ΔC: ALIX lacking the C-terminus; BRO1: Bro1-domain; ΔBRO1: ALIX lacking the Bro1-domain. **d**, **e** Yeast two-hybrid analysis between GAD-fused CaLB1 and GBD-fused ALIX. Results of three independent transformants are shown. -LW: media without leucine and tryptophan, -LWH: media without leucine, tryptophan, and histidine, 3-AT: 3-Amino-1,2,4-triazole. The experiment was conducted three times, and one representative result is shown. **f** in vitro binding assay between CaLB1-6xHis and MBP-ALIX. MBP-fused ALIX (BRO1) and ALIX(ΔBRO1) were mixed with CaLB1-6xHis bound to TALON beads. After incubation beads were washed and bound proteins were eluted and subjected to immunoblots using an anti-MBP antibody and an anti-His antibody. The experiment was conducted three times, and one representative result is shown. **g** in vitro binding assay between CaLB1(C2)−6xHis or CaLB1 (FL)−6xHis and GST-ALIX. His-tag fused CaLB1 proteins were mixed with GST-ALIX bound to glutathione Sepharose. After incubation and extensive washing, bead-retained proteins were eluted and subjected to immunoblots using an anti-GST antibody and an anti-His antibody. The experiment was conducted twice, and one representative result is shown. **h** Root epidermal cells of seedlings expressing both *ALIXpro: GFP-ALIX* and *CaLB1pro: CaLB1-mRFP* were analyzed under a confocal microscope. 485 CaLB1-mRFP foci from 17 images were analyzed and 35.3% of the mRFP-positive showed colocalization with GFP-ALIX. Scale bars: 10 μm. The magnified region is indicated with a white rectangle. Scale bar in the magnification: 1 μm. **a**, **f**, **g** Source data are provided as a Source Data file.

or presence of the phosphatidylinositol 3-kinase inhibitor Wortmannin (WM), respectively (Supplementary Fig. 3b). These results suggest that CaLB1 does not constantly localize to endosomes but may associate with a higher frequency when trafficking is inhibited at the late endosomes.

Another cellular organelle enriched in PI(3)P is the autophagosome[27]. To establish a possible function of CaLB1 on the autophagosomes, we tested whether CaLB1 interacts with ATG8. Arabidopsis has nine ATG8 homologues[31]. An in vitro pull-down assay with CaLB1-GST and His-tagged ATG8a, ATG8f, and ATG8i, representing Clade I (ATG8a and ATG8f) and Clade II (ATG8i) ATG8s, showed that CaLB1-GST can directly interact with His-tagged ATG8s (Fig. 3a and Supplementary Fig. 3c). As ATG8 is a ubiquitin-like protein, we also tested whether CaLB1 interacts with ubiquitin. MST analyses of fluorescently labeled CaLB1(C2) with ubiquitin and ATG8i showed that CaLB1 interacts with ATG8 but not with ubiquitin (Fig. 3b and Supplementary Fig. 3d).

We next searched for known linear motifs for ATG8 interaction, [W/F/Y]XX[V/L/I], or ATG8-interacting motifs (AIM)[31] in CaLB1 (Fig. 3c and Supplementary Fig. 3e). Using the iLIR web database[32], we identified four putative AIMs (Fig. 3c and Supplementary Fig. 3e) and investigated whether substitutions in the AIM sequences have an impact on the ATG8-CaLB1 interaction using Y2H assay. While GAD-CaLB1(Y172A) and GAD-CaLB1(Y205A) still interacted with GBD-ATG8i, GAD-CaLB1(Y34A) and GAD-CaLB1(F64A) did not bind GBD-ATG8i (Fig. 3c and Supplementary Fig. 3e, f), suggesting that these amino acid residues are important for the interaction between CaLB1 and ATG8s.

### *calb1* mutants are sensitive to salt

Autophagy is known to be upregulated by nutrient starvation, drought, heat, and salt stress[33–36]. We therefore investigated whether CaLB1 and ALIX could function in autophagy under specific stress conditions. An ePLANT[37] search (Abiotic Stress eFP at bar.utoronto.ca/eplant) indicated that the transcription of both *ALIX* and *CaLB1* is induced by 150 mM NaCl. As salt stress was reported to rapidly induce autophagy[36,38], we analyzed the expression of *CaLB1, ALIX*, and *ATG8s* upon salt treatment over time. Quantitative real time (qRT)-PCR

Mammalian ALIX was reported to interact with lysobisphosphatidic acids (LBPA)[25] that are not produced in plants and fungi[26]. To test the binding of Arabidopsis ALIX to phospholipids, we prepared recombinant Arabidopsis ALIX along with its yeast and human orthologs ScBro1 and HsALIX, respectively, and performed a lipid overlay assay (Fig. 2e and Supplementary Fig. 2h). HsALIX did not bind any of the phospholipids used in the assay whereas Arabidopsis ALIX and Bro1 interacted with PI(3)P. These results indicate that both CaLB1 and Arabidopsis ALIX can bind cellular membranes containing PI(3)P.

### CaLB1 interacts with the autophagosomal component ATG8

PI(3)P is a phosphoinositide enriched in endosomes and autophagosomes in plants[27]. Since ALIX was previously shown to localize on late endosomes[13,14], we first investigated whether CaLB1 is also localized on endosomes. For this, Arabidopsis seedlings co-expressing *CaLB1pro: CaLB1-GFP* and the early endosomal marker *SYP43pro:mRFP-SYP43*[28] or the late endosomal marker *ARA6pro: ARA6-mRFP*[29] were generated. 7.8% and 6.6% of CaLB1-GFP foci colocalized with mRFP-SYP43 in the absence or presence of the ARF-GEF inhibitor Brefeldin A (BFA), respectively[30] (Supplementary Fig. 3a). On the other hand, 3.5% and 16.6% of CaLB1-GFP foci co-localized with ARA6-mRFP in the absence

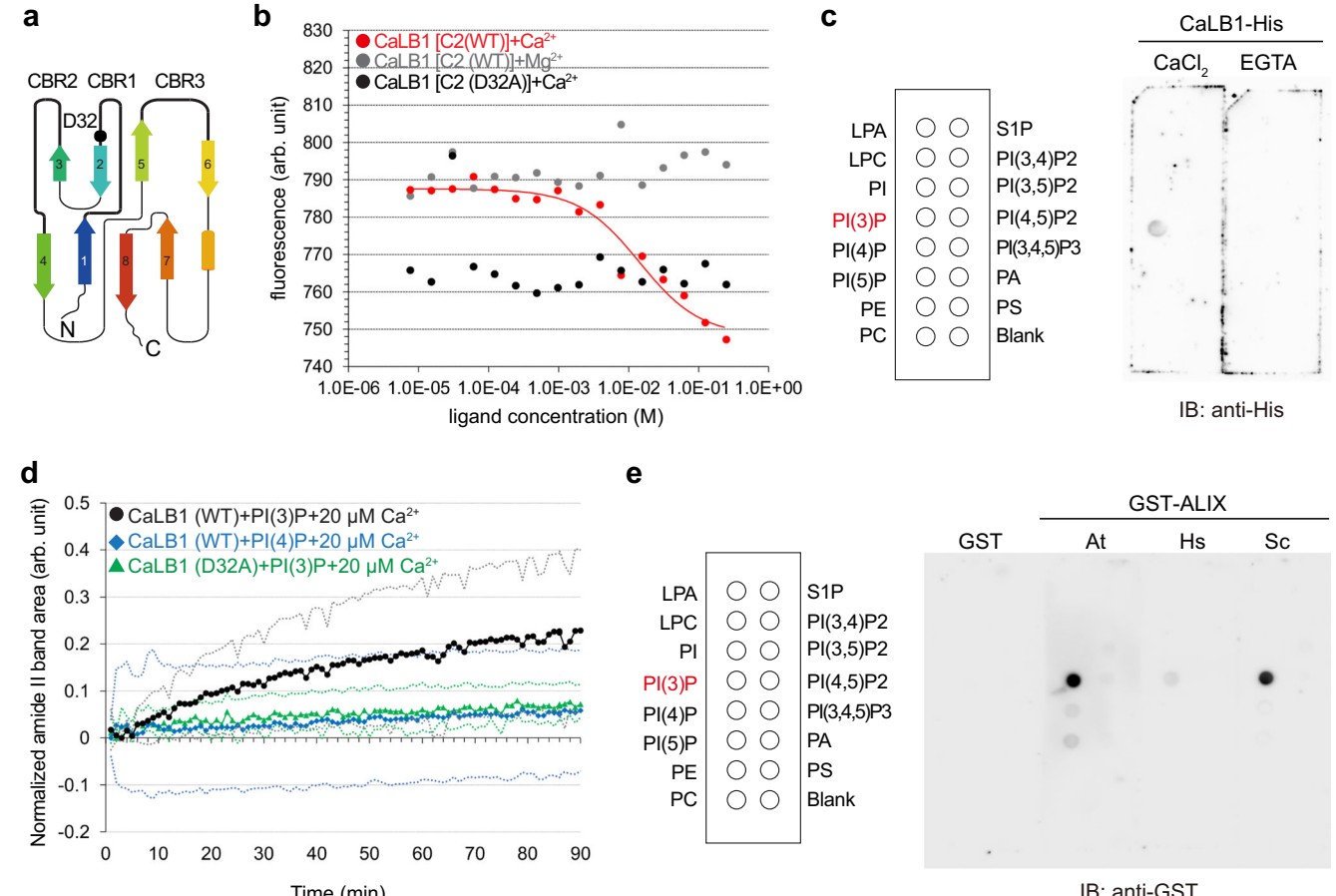

**Fig. 2 | CaLB1 and ALIX bind the anionic phospholipid PI(3)P. a** Organization of the C2 domain of CaLB1. D32 indicates the mutation introduced in the putative Ca²⁺-binding region CBR1. CBR: Ca²⁺-binding region; N: N-terminal; C: C-terminal. **b** MicroScale Thermophoresis analysis of the wild-type or D32A-mutated C2-domain of CaLB1 with either Ca²⁺ or Mg²⁺. 50 nM of labeled recombinant CaLB1(C2) was mixed with buffers containing either Ca²⁺ or Mg²⁺ at a sequential dilution starting from 250 mM. Two independent measurements were performed, and one representative result is shown. **c** Lipid overlay assays with recombinant CaLB1-6xHis in the presence of 10 μM CaCl₂ or 5 mM EGTA. Bound proteins were detected with an anti-His antibody. Lipid species on the membrane are indicated on the left. LPA Lysophosphatidic acid, LPC Lysophosphatidylcholine, PI Phosphatidylinositol, PE Phosphatidylethanolamine, PC Phosphatidylcholine, S1P Sphingosine 1-P, PA

Phosphatidic acid, PS Phosphatidylserine. The experiment was conducted three times, and one representative result is shown. **d** Recombinant wild-type (WT) or mutated (D32A) CaLB1 proteins were incubated with solid-supported lipid bilayers containing consisting of POPC supplemented with 1 % PI(3)P or PI(4)P in the presence of 20 μM of Ca²⁺. The integrated amide II band area of CaLB1 was measured using ATR-FTIR spectroscopy. Average of the measurements was plotted from 3 independent experiments. Dotted lines indicate the upper and lower limit of standard deviations. **e** Lipid overlay assays with GST-fusions of ALIX orthologs from Arabidopsis, human, and yeast (Bro1p) in the presence of 10 μM CaCl₂. Bound proteins were detected with an anti-GST antibody. At: *Arabidopsis thaliana*, Hs: *Homo sapiens*, Sc: *Saccharomyces cerevisiae*. The experiment was conducted twice and one of the results is shown. **b**, **c**, **d**, **e** Source data are provided as a Source Data file.

analyses showed that the expression of *CaLB1*, but not *ALIX* or the *ATG8s*, significantly increased after 30 min of salt treatment (Fig. 4a and Supplementary Fig. 4a). Seedlings expressing either *CaLB1-GFP*, *GFP-ALIX*, or *GFP-ATG8a* showed an increased amount of GFP-fusion protein levels upon salt treatment (Fig. 4b), suggesting that, in addition to transcriptional regulation of CaLB1, there are posttranscriptional mechanisms also regulating ALIX and ATG8a upon NaCl treatment.

We then generated CRISPR mutants of *CaLB1* to analyze the function of CaLB1 in autophagy. Two independent *calb1* lines were isolated and named *calb1-1* and *calb1-2*. Both lines have indel mutations in the first exon that generates a premature stop codon and show reduction in *CaLB1* transcripts probably due to nonsense-mediated decay[39] (Supplementary Fig. 4b–e). We first investigated whether CaLB1 function is required for carbon starvation-induced autophagy. To test this, we subjected wild-type, *calb1-1*, *calb1-2*, and *atg10-1* seedlings to prolonged dark treatment. In contrast to the autophagy mutant *atg10-1*, *calb1-1*, and *calb1-2* mutants did not show a significant reduction in

chlorophyll contents when compared with the wild type (Supplementary Fig. 4f, g), suggesting that CaLB1 is not essential for carbon starvation-induced autophagy.

If CaLB1 function is necessary for proper salt-response, *calb1* mutants would show altered response to NaCl treatment. We therefore subjected wild-type, *calb1-1*, and *calb1-2* seedlings together with *atg10-1* mutants to media containing 100 mM NaCl. Similar to *atg10-1* seedlings, *calb1-1*, *calb1-2*, showed a significant reduction in primary root length compared with the wild type (Fig. 4c, d and Supplementary Fig. 4h–j), suggesting that CaLB1 is required for proper salt response in seedlings. The salt sensitivity of *calb1-1* was complemented with a *CaLB1pro: CaLB1-GFP* construct (Fig. 4c–e and Supplementary Fig. 4h), showing that the CaLB1-GFP fusion protein is functional and that the NaCl-sensitive phenotype is due to the mutation in the *CaLB1* locus.

### CaLB1 and ALIX localize on salt-induced autophagosomes

We next addressed whether the regulation of CaLB1 and ALIX by NaCl is accompanied with a change in their intracellular localization. First, the number of CaLB1 and ALIX foci as well as the number of

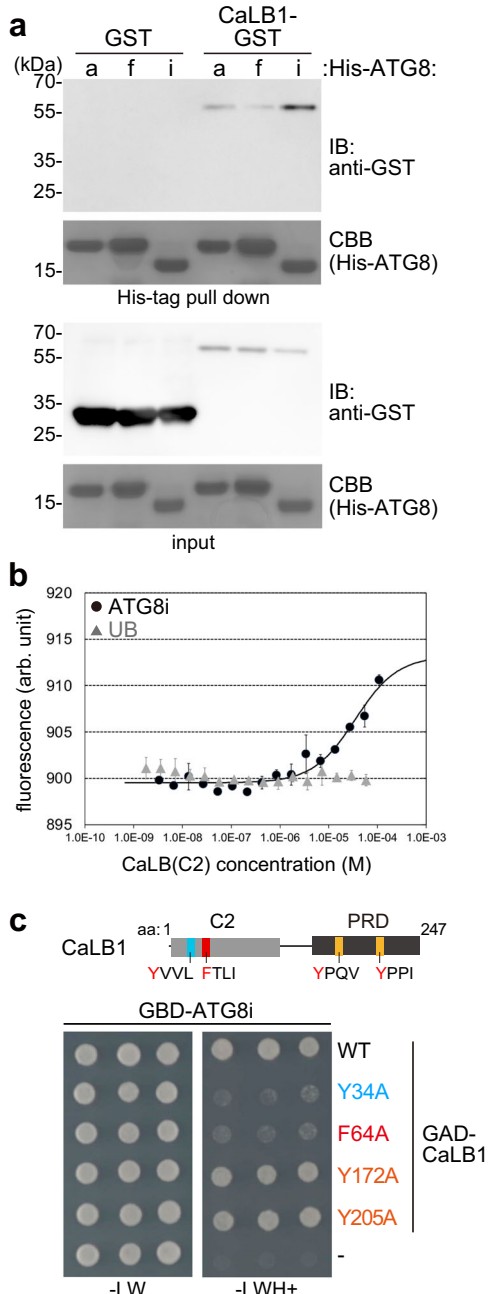

**Fig. 3 | CaLB1 interacts with ATG8. a** In vitro binding assays of GST and CaLB1-GST with His-tagged ATG8s bound to TALON beads. Bound proteins were analyzed with an anti-GST antibody. His-ATG8s were visualized by protein staining using CBB. The experiment was conducted twice, and one representative result is shown. **b** MicroScale Thermophoresis analysis of the C2 domain of CaLB1 with ATG8i or ubiquitin (UB). The changes in fluorescence are shown in relation to the concentration of ATG8i or UB. The experiment was conducted twice, each time with three technical replicates. One representative result is shown. Error bars: standard deviation. **c** Yeast two-hybrid analysis of the regions in CaLB1 responsible for the interaction with ATG8i. Four aromatic amino acids (Y34, F64, Y172, and Y205, positions shown in the scheme) were each mutated and the impact of the mutation was tested in a yeast two-hybrid assay. PRD proline-rich domain, -LW growth media lacking leucine and tryptophan, -LWH growth media lacking leucine, tryptophan, and histidine, 3-AT 3-Amino-1,2,4-triazole. The experiment was conducted three times, and one representative result is shown. **a, b** Source data are provided as a Source Data file.

GFP-ATG8a-marked autophagosomes were analyzed in seedlings expressing either *CaLB1pro: CaLB1-GFP*, *ALIXpro: GFP-ALIX*, or *UBQ10-pro: GFP-ATG8a*. Upon 150 mM NaCl treatment, the number of GFP-ATG8a-marked autophagosomes increased as reported[36], but also the number of CaLB1-GFP and GFP-ALIX foci increased significantly (Fig. 5a−d) and the colocalization efficiency of CaLB1-mRFP with GFP-ALIX increased to 54.0% (Fig. 5e).

Since CaLB1 binds PI(3)P-containing membranes in a $Ca^{2+}$-dependent manner, we further tested whether the localization of CaLB1 is altered when cytosolic levels of $Ca^{2+}$ is altered. For this purpose, we treated CaLB1-GFP expressing seedlings with 150 μM Lanthanum(III) chloride ($LaCl_3$), a salt that blocks the activity of divalent cation channels with a high affinity for $Ca^{2+}$ channels[40]. The number of CaLB1-GFP positive intracellular compartments did not change significantly upon $LaCl_3$ treatment (Supplementary Fig. 4k, l), showing that a bulk change in cytosolic $Ca^{2+}$ does not affect the localization of CaLB1 in vivo.

We subsequently investigated whether the CaLB1- and ALIX-positive foci represent autophagosomes. For this end, we generated a transgenic Arabidopsis line expressing *CaLB1pro: CaLB1-GFP* together with the autophagosome marker *UBQ10pro:mRFP-ATG8i*. Upon 2 h of treatment with 150 mM of NaCl, CaLB1-GFP localized on a defined region on the rim of the mRFP-ATG8i-marked autophagic membrane (Fig. 5f), and occasionally also at the plasma membrane (Supplementary Fig. 5a). To test whether the localization on the autophagic membranes differs between the ATG8 clades[31], we generated Arabidopsis lines expressing *UBQ10pro:mRFP-ATG8a* or *UBQ10pro:mRFP-ATG8e* in addition to mRFP-ATG8i-expressing lines. CaLB1-GFP also localized on the autophagosomal membranes marked with mRFP-ATG8a or mRFP-ATG8e, indicating that the colocalization between CaLB1 and ATG8s does not depend on specific ATG8 clades (Fig. 5f, k and Supplementary Fig. 5b−e). 75% of the autophagosomes positive for CaLB1 had a single CaLB1 puncta, while 22.2% and 2.8% had two and three CaLB1-GFP foci, respectively (Fig.5g and Supplementary Movie 1 and 2). In 69.4% of the autophagosomes, CaLB1-GFP was stably associated with autophagic membranes. However, association with and disassociation from the autophagosomes could also be observed over time. In 8.3% and 11.1% of the autophagosomes, the CaLB1-GFP puncta appeared at or disappeared from the autophagosomes, respectively (Supplementary Fig. 5c, d and Supplementary Movie 3).

To analyze the localization of CaLB1 at the ultrastructure level, we performed immunogold labelling on *CaLB1pro: CaLB1-GFP* expressing seedlings. Anti-GFP antibodies were found to label autophagic membranes (Fig. 5h), confirming that CaLB1 associated with autophagosomes. To investigate whether CaLB1-GFP is present in membrane fractions enriched in autophagosomes, we next carried out a multi-step fractionation[38] using total extracts prepared from NaCl-treated CaLB1-GFP expressing seedlings. Immunoblot analysis revealed that CaLB1 is found in the autophagosome-enriched fractions together with ATG8 and NBR1 (Fig. 5i).

We next wanted to understand whether the localization of CaLB1 on autophagosomes requires the interaction between CaLB1 and ATG8. For this, we generated transgenic Arabidopsis lines expressing *CaLB1pro: CaLB1(F64A)-GFP* with *UBQ10pro:mRFP-ATG8i*. Whereas the wild-type CaLB1 colocalized with autophagosomes, colocalization between CaLB1(F64A)-GFP, a variant not interacting with ATG8, and mRFP-ATG8i was reduced when compared to that of the wild-type CaLB1 (Fig. 5j, k and Supplementary Fig. 3e). This results suggest that the interaction between CaLB1 and ATG8 is important for the recruitment of CaLB1 to autophagosomes.

We next analyzed Arabidopsis seedlings co-expressing *ALIXpro: GFP-ALIX* and *UBQ10pro:mRFP-ATG8i* to establish whether ALIX also

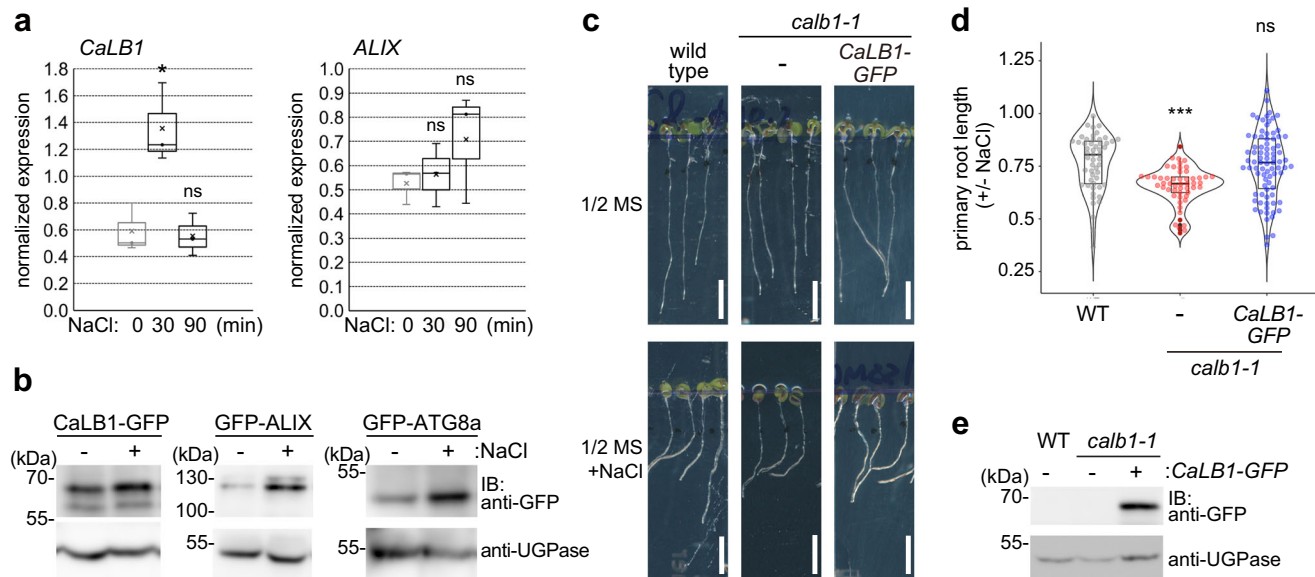

**Fig. 4 | CRISPR mutants of *CaLB1* show salt sensitivity. a** Expression of *CaLB1* and *ALIX* after 30- or 90 min treatment with 150 mM NaCl normalized against *ACTIN8*. Both plots show the results of three experiments each of which is the average of technical quadruplicates. For *CaLB1*: NaCl 0 min/30 min $p = 0.0270$ (*0.01 < $p$ < 0.05), NaCl 0 min/90 min $p = 0.816$ (ns: not significant, $p > 0.05$), for *ALIX*: NaCl 0 min/30 min $p = 0.692$ (ns not significant), NaCl 0 min/90 min $p = 0.303$ (ns). **b** Seedlings containing *CaLB1pro:CaLB1-GFP*, *ALIXpro:ALIX-GFP*, or *UBQ10-pro:GFP-ATG8a* were treated for 2 h with 150 mM NaCl. Immunoblotting was performed using an anti-GFP antibody. An anti-UGPase antibody was used for loading control. The experiment was conducted three times, and one representative result is shown. **c** Photographs of 7-day-old wild-type, *calb1-1*, and *calb1-1* expressing *CaLB1pro: CaLB1-GFP* seedlings were transferred after three days to media with or without 100 mM NaCl and were grown for further 4 days. Scale bars: 5 mm. **d** Ratio of primary root length with and without NaCl treatment as in (**c**). Wild type ($n = 53$ and 49 for +/− NaCl), *calb1-1* ($n = 53$ and 65 for +/− NaCl), *calb1-1/CaLB1-GFP* ($n = 100$ for both +/− NaCl). The root length after NaCl treatment was divided by the average root length of untreated seedlings. Wild type/*calb1-1* $p = 1.83 \times 10^{-6}$ (***$p < 0.001$), wild type/*calb1* with *CaLB1-GFP* $p = 0.734$ (ns). The experiment was conducted three times, and one representative result is shown. **e** Total extracts were prepared from 7-day-old seedlings as in (**c**) and subjected to immunoblotting with an anti-GFP antibody. An anti-UGPase antibody was used for loading control. The experiment was conducted three times, and one representative result is shown. **a**, **b**, **d**, **e** Source data are provided as a Source Data file. **a**, **d** Box plot: center line, median; box limits, first and third quartiles; whiskers, 1.5x interquartile range Two-tailed *t*-test, no equal variance.

localizes on autophagosomes. Like CaLB1, GFP-ALIX localized on a defined region of salt-induced autophagic membranes (Fig. 5l). In 69.6% of the analyzed autophagosomes, GFP-ALIX was found as a single punctum on the mRFP-marked autophagosome, while 17% and 5.6% of the autophagosomes had two and three foci, respectively. In contrast to CaLB1-GFP, in 9.0% of the autophagosomes, GFP-ALIX showed a ring-like distribution along the mRFP-ATG8-marked autophagosomal membrane (Fig. 5m), suggesting that CaLB1 and ALIX can localize to autophagosomes by different mechanisms.

The number of ALIX puncta on salt-induced autophagosomes (25.7%, $n = 1036$ autophagosomes) was comparable to that under non-stress conditions (27.9%, $n = 408$ autophagosomes) (Supplementary Fig. 5f). The colocalization efficiency between CaLB1 and ATG8i slightly increased upon salt treatment (11.6%, $n = 250$ autophagosomes) (Supplementary Fig. 5g). This suggests that CaLB1 and ALIX could be recruited to autophagosomes independently from specific stresses. However, since the number of autophagosomes increases upon salt treatment (Fig. 5b, d), more autophagosomes are positive for CaLB1 and ALIX under salt stress conditions.

### The IDR of CaLB1 induces assembly of CaLB1 and ALIX into condensates

The round CaLB1-GFP foci resembled typical phase-separated biomolecular condensates. We thus tested whether they can be dissolved by treatment with alcohol that interferes with the hydrophobic interactions between molecules in condensates[41–44]. 5-day-old seedlings expressing *CaLB1pro:CaLB1-GFP* were treated with 5% of 1,6-hexanediol, an organic compound used to dissolve molecular condensates. As prolonged treatment with 1,6-hexanediol was reported to cause various side effects[45], we analyzed the seedlings immediately

after treatment. The number of CaLB1-GFP puncta decreased significantly upon 5% 1,6-hexanediol treatment, whereas the less hydrophobic 1,2,6-hexanetriol did so to a lesser extent (Fig. 6a, b), suggesting that CaLB1-GFP can form condensates in vivo.

We next tested whether 1,6-hexanediol impact CaLB1 and ALIX foci upon salt treatment. For this, we treated seedlings expressing either CaLB1-GFP or GFP-ALIX together with mRFP-ATG8i for 2 h with 150 mM NaCl, followed by treatment with 5% 1,6-hexanediol (Fig. 6c–f). The number of CaLB1 puncta decreased significantly upon 1,6-hexanediol treatment, whereas the number of autophagosomes marked with mRFP-ATG8i did not (Fig. 6c, e). 15 min after removal of 1,6-hexanediol, CaLB1-GFP foci reappeared (Supplementary Fig. 6a), suggesting that CaLB1 can dynamically move between the cytosol and condensates. Like CaLB1, the number of GFP-ALIX-containing foci also decreased significantly upon 1,6-hexanediol treatment, while mRFP-ATG8i-marked autophagosomes were resistant to the treatment (Fig. 6d, f). These results suggest that CaLB1 and ALIX localize in phase-separated condensates on autophagosomes in plants cells.

In a recent publication, human ALIX was shown undergo phase separation and that the PRD is important for the condensate formation[46]. To understand whether the presence of CaLB1 influences the formation of Arabidopsis ALIX condensates, we analyzed the number of GFP-ALIX puncta in Arabidopsis root cell culture-derived protoplasts. We generated CRISPR(CaLB1) and CRISPR(mutCaLB1) constructs that express the Cas9 nuclease together with either a functional or a mutated guide RNA for *CaLB1*, respectively (Fig. 6g). When expressed in protoplasts, the CRISPR(CaLB1) construct reduced the CaLB1-mRFP protein levels compared with the control or with the CRISPR(mutCaLB1) construct (Fig. 6h). The co-expression of CRISPR(CaLB1), but not CRISPR(mutCaLB1), caused a decrease in the

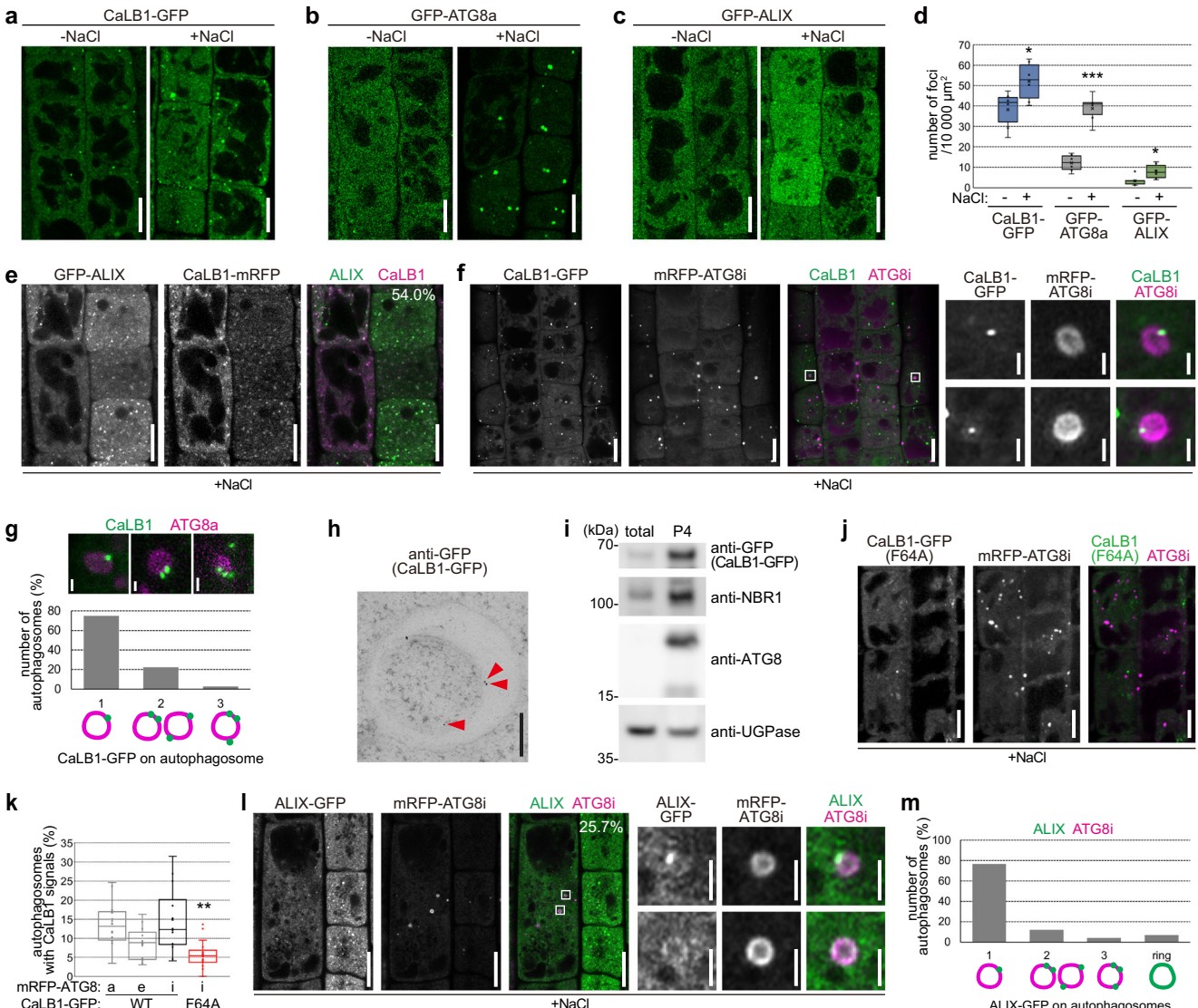

**Fig. 5 | CaLB1 and ALIX localize on salt-induced autophagosomes.** Confocal microscopy of NaCl-treated root epidermal cells expressing *CaLB1pro: CaLB1-GFP* (**a**), *UBQ10pro: GFP-ATG8a* (**b**), or *ALIXpro: GFP-ALIX* (**c**). Scale bars: 10 μm. The experiment was conducted six times, and one representative image is shown. **d** Quantification of (**a**–**c**). CaLB1-GFP −/+ NaCl $p = 0.0289$ (*$0.01 < p < 0.05$), GFP-ATG8a −/+ NaCl $p = 2.32 \times 10^{-5}$ (***$p < 0.001$), ALIX-GFP −/+ NaCl $p = 0.0370$ (*$0.01 < p < 0.05$). The analysis was conducted six times, and the average of all experiments is shown. Seedlings expressing *ALIXpro: GFP-ALIX* and *CaLB1pro: CaLB1-mRFP* (**e**) or *CaLB1pro: CaLB1-mRFP* and *UBQ10pro:mRFP-ATG8i* (**f**) were treated with 150 mM NaCl for 2 h before confocal imaging. 54% of CaLB1 foci ($n = 506$ from 12 seedlings) were positive for the GFP-ALIX. Scale bars: 10 μm. **g** Analysis of time-lapse images of NaCl-treated *CaLB1pro:CaLB1-GFP/UBQ10pro:mRFP-ATG8a* seedlings. $n = 36$ autophagosomes (16 seedlings). **h** Immuno-electron microscopy on CaLB1-GFP using an anti-GFP antibody after treatment with 150 mM NaCl for 2 h. Arrowhead: gold particle. Scale bar: 200 nm. Three seedling roots were analyzed, and one representative image is shown. **i** Membrane fraction (P4) was analyzed by immunoblotting along with the total extract using anti-GFP, anti-NBR1, anti-ATG8, and anti-UGPase antibodies. **j** Seedlings expressing *CaLB1pro:*

*CaLB1(F64A)-GFP* and *UBQ10pro:mRFP-ATG8i* were treated with 150 mM NaCl for 2 h before confocal imaging. Scale bars 10 μm. **k** Number of ATG8a-, ATG8e-, or ATG8i foci with CaLB1-GFP or CaLB1-GFP(F64A) signals. ATG8i with CaLB1-GFP/CaLB1-GFP(C2mut) $p = 0.00416$ (**$0.001 < p < 0.01$). The analysis was conducted twice, and all results are shown. For CaLB1-GFP, $n = 728$ autophagosomes in 14 seedlings for ATG8a, $n = 506$ (15 seedlings) for ATG8e, $n = 586$ (12 seedlings) for ATG8i, and for CaLB1-GFP(F64A) and ATG8i $n = 1150$ (24 seedlings) were analyzed. **l** Seedlings expressing *ALIXpro: GFP-ALIX* and *UBQ10pro:mRFP-ATG8i* were treated with 150 mM NaCl for 2 h before confocal imaging. scale bars: 10 μm. 25.7% of the mRFP-ATG8i marked autophagosomes ($n = 1036$ from 11 roots) were positive for GFP-ALIX signals. Rectangle: magnified region. Scale bar in the magnification: 1 μm. **m** Classification of GFP-ALIX signals on autophagosomes. Images taken as in (**l**) were analyzed ($n = 11$ roots). **d**, **g**, **k**, **l**, **m** Source data are provided as a Source Data file. **d**, **k** Box plot: center line, median; box limits, first and third quartiles; whiskers, 1.5x interquartile range. Two-tailed *t*-test, no equal variance. **e**, **f**, **i**, **j**, **l** Experiments were conducted three times and one representative result is shown.

number of GFP-ALIX foci (Fig. 6i, j), though GFP-ALIX was expressed at comparable level in both samples (Supplementary Fig. 6b). We performed the experiment using protoplasts isolated from roots of wild-type and *calb1-1* seedlings. Similarly, reduced numbers of GFP-ALIX puncta were observed in protoplasts derived from *calb1-1* seedling roots compared to the wild type (Supplementary Fig. 6c–e). These

results suggest that the presence of CaLB1 enhanced the puncta formation of ALIX in vivo.

Low complexity regions or intrinsically disordered regions (IDRs) in proteins are known to serve as the main drivers for phase separation and the formation of condensates[47]. Analysis of the amino acid sequence using IUPred2[48] (https://iupred2a.elte.hu) and AlphaFold[49]

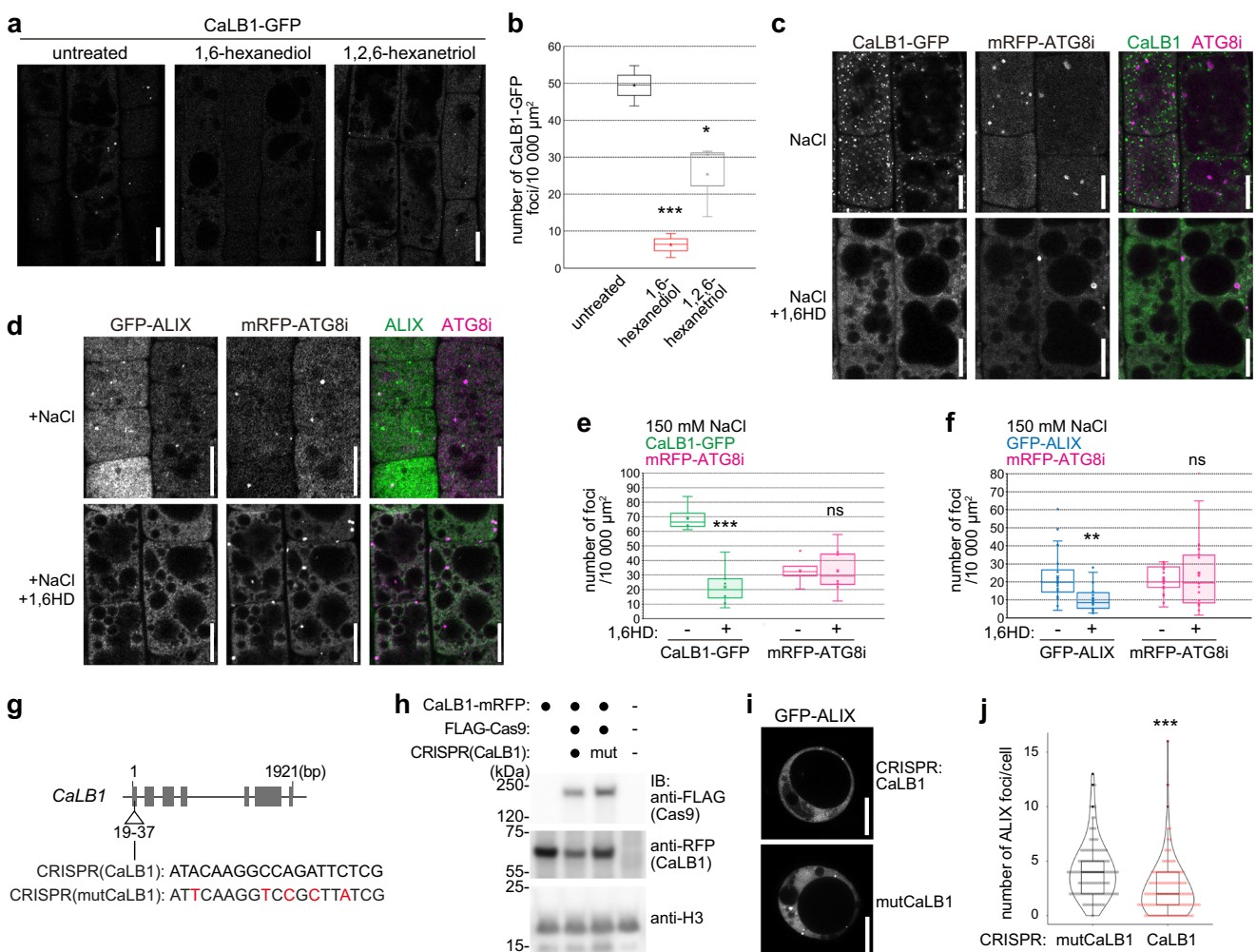

**Fig. 6 | CaLB1 foci are resolved upon 1,6-hexanediol treatment. a** Localization of CaLB1-GFP under the native promoter in root epidermis cells treated with 1,6-hexanediol or 1,2,6-hexanetriol. The experiments were conducted three times, and representative images are shown. Scale bars: 10 μm. **b** Number of CaLB1-GFP-positive foci in (**a**). The experiments were conducted three times, and the mean for each experiment is shown [($n = 23$ seedlings (untreated); $n = 26$ seedlings (1,6-hexanediol); $n = 28$ seedlings (1,2,6-hexanetriol)). untreated/1,6-hexandiol $p = 0.000892$ (***$p < 0.001$), untreated/1,2,6-hexanetriol $p = 0.0333$ (*$0.01 < p < 0.05$). Seedlings co-expressing *UBQ10pro:mRFP-ATG8i* and *CaLB1pro: CaLB1-GFP* (**c**) or *ALIXpro: GFP-ALIX* (**d**) were analyzed under a confocal microscope upon NaCl or NaCl and 1,6-hexanediol (1,6-HD) treatment. The experiment with CaLB1 and ALIX were conducted two and three times, respectively, and representative images are shown. Scale bars: 10 μm. **e**, **f** Quantification of the results in (**c**) and (**d**). $p = 1.50 \times 10^{-4}$, ***$p < 0.001$ (CaLB1-GFP) and $p = 0.00186$, **$0.001 < p < 0.01$ (GFP-ALIX), $p = 0.998$ (**c**, ATG8i) and 0.430 (**d**, ATG8i), not significant (ns, $p > 0.05$)].

In total, for +NaCl/+NaCl+1,6-HD, 249/128 foci in 4 roots were analyzed for CaLB1-GFP and ATG8i, 178/269 foci in 8 roots for CaLB1-GFP and ATG8i, for GFP-ALIX and ATG8i, 485/477 foci in 8 roots, and for GFP-ALIX and ATG8i, 220/570 foci in 8 roots. **g** CRISPR target [CRISPR(CaLB1)] and mutated target [CRISPR(mutCaLB)] sequences. **h** Protein extracts of protoplasts transformed with *CaLB1pro:mRFP-CaLB1* alone or with *CaLB1pro:mRFP-CaLB1* and *CRISPR(CaLB1)/3xFLAG-Cas9* or *CRISPR(mutCaLB)/3xFLAG-Cas9* were subjected to anti-FLAG and anti-RFP immunoblotting. Histon H3 was used as loading control. The experiment was conducted twice, and one representative result is shown. **i** Representative images of Arabidopsis root cell culture-derived protoplasts transformed with the CRISPR constructs shown in (**g**) co-transformed with GFP-ALIX. Scale bar: 10 μm. **j** Violin plot of the number of GFP-ALIX-positive foci per cell in (**i**) [$n = 120$ *CRISPR(CaLB1)*, $n = 111$ *CRISPR* (mutCaLB1)]. *CRISPR(CaLB1)/CRISPR(mutCaLB1)*: $p = 5.69*10^{-7}$ (***$p < 0.001$). (**b**, **e**, **f**, **h**, **j**) Source data are provided as a Source Data file. **b**, **e**, **f**, **j** Box plot: center line, median; box limits, first and third quartiles; whiskers, 1.5x interquartile range. Two-tailed *t*-test, no equal variance.

modelling identified an IDR at the C-terminus of CaLB1 and ALIX (Fig. 7a). To test whether the IDR of CaLB1 is required for phase separation into condensates and whether it can trigger the formation of ALIX condensates, we prepared recombinant CaLB1-6xHis, CaLB1(C2)−6xHis, and ALIX. CaLB1 and CaLB1(C2) were labelled with NHS-Red and ALIX was labelled with Alexa Fluor 488 and subsequently formation of condensates in vitro was analyzed. After 30 min of incubation, we found that full-length CaLB1 and ALIX formed round structures, whereas CaLB1(C2) did not (Fig. 7b and Supplementary Fig. 7a, b), indicating that the C-terminal IDR of CaLB1 is necessary for the assembly of CaLB1 into condensates.

The number of ALIX-containing condensates increased by increasing concentrations of CaLB1, whereas the number of CaLB1-containing condensates was not dependent on the concentration of ALIX (Fig. 7c), suggesting that the presence of CaLB1 enhances the condensate formation of ALIX. When the IDR of CaLB1 was missing, the number of ALIX condensates in the mixture was reduced (Fig. 7c), suggesting that the interaction between CaLB1 and ALIX triggers their assembly into condensates by interaction through their PRDs. The size of both ALIX and CaLB1 condensates increased in a concentration dependent manner and the IDR of CaLB1 was required for this effect (Fig. 7d).

We subsequently analyzed the mobility of the labeled proteins in condensates using fluorescent recovery after photobleaching (FRAP) analyses. Recovery of fluorescent signals was observed for both CaLB1 and ALIX; however, ALIX was less dynamic compared to CaLB1, and the

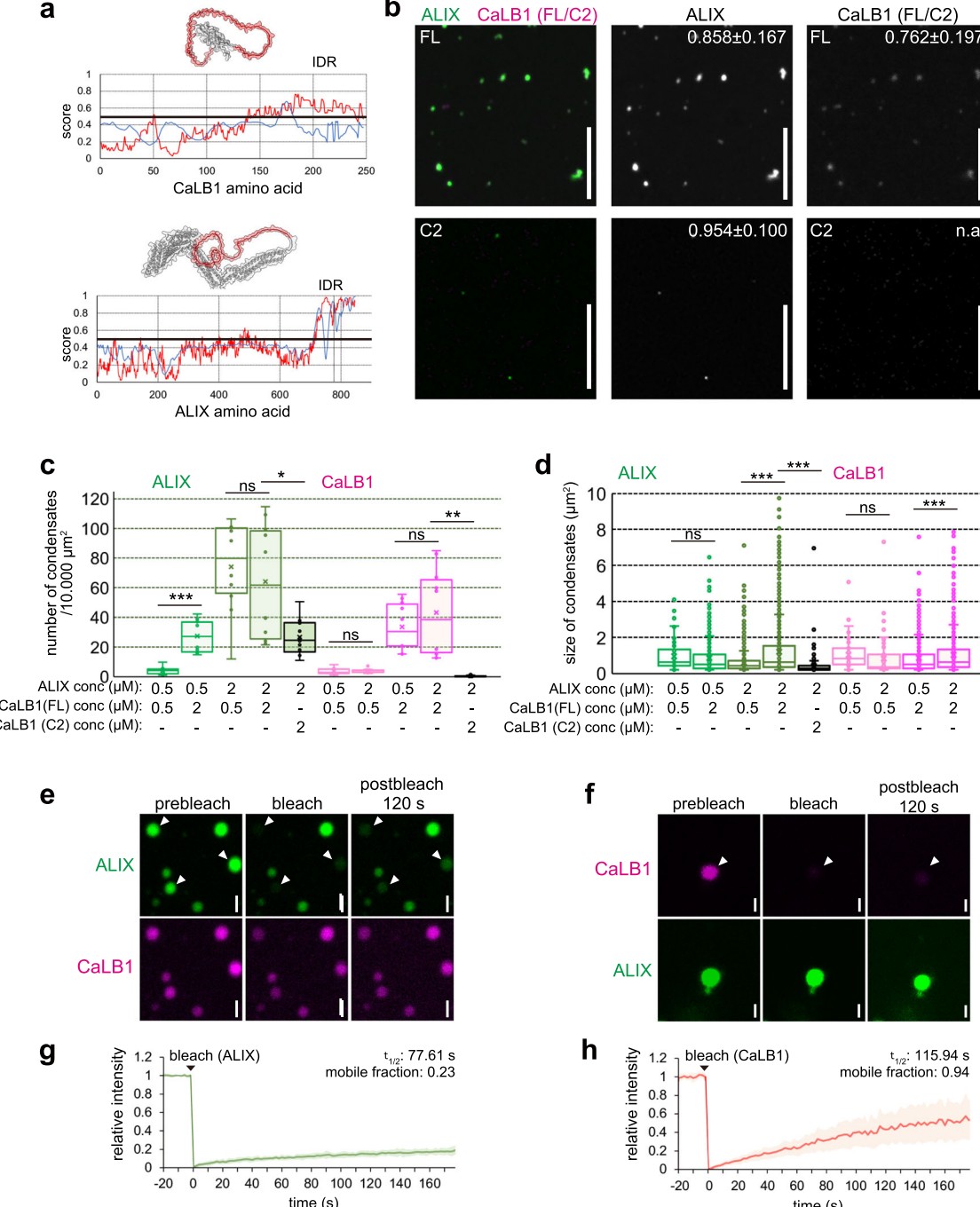

**Fig. 7 | CaLB1 and ALIX assemble into condensates. a** Predicted IDRs are highlighted (red) in the AlphaFold models. red line, IUPred2 (long); gray line, ANCHOR2. **b** 2 μM of labelled ALIX was mixed with 2 μM of labelled CaLB1 (FL) or CaLB1(C2) to analyze condensate formation. The experiment was conducted twice, and one representative image is shown. Mean circularity±standard deviation is shown. For ALIX/CaLB1(FL), 488 and 323 condensates were measured, respectively. In ALIX/CaLB1(C2), only ALIX condensates (*n* = 397) were measured as the number of CaLB1(C2) condensates were low. Scale bars: 10 μm. Number (**c**) and size (**d**) of condensates in two experiments (*n* = 10 images), except for the combination CaLB1(C2) 2 μM + ALIX 2 μM for which 13 images from two experiments were analyzed. All results are shown. **c** *p*-values for 0.5 μM ALIX, CaLB1 0.5 μM/2 μM *p* = 7.89×10⁻⁵ (***p < 0.001); for 2 μM ALIX, CaLB1(FL) 0.5 μM/2 μM *p* = 0.537(ns: not significant); for 2 μM ALIX, CaLB1(FL) 2 μM/CaLB1(C2) 2 μM *p* = 0.0158 (*0.01 < p < 0.05); for CaLB1(FL) 0.5 μM, ALIX 0.5 μM/2 μM *p* = 0.658 (ns); for CaLB1(FL) 2 μM, ALIX 0.5 μM/2 μM *p* = 0.402 (ns); for CaLB1(FL) or CaLB1(C2) 2 μM

with ALIX 2 μM; CaLB1(FL)/CaLB1(C2) *p* = 0.00151 (**0.001 < p < 0.01). **d** *p*-values for 0.5 μM ALIX, CaLB1 0.5 μM/2 μM *p* = 0.285 (ns); for 2 μM ALIX, CaLB1 0.5 μM/2 μM *p* = 5.98 × 10⁻⁴⁰(***p < 0.001); for 2 μM ALIX, CaLB1 2 μM/CaLB1(C2) 2 μM *p* = 2.23 × 10⁻⁷² (***p < 0.001); for CaLB1 0.5 μM, ALIX 0.5 μM/2 μM *p* = 0.174 (ns); for CaLB1 2 μM, ALIX 0.5 μM/2 μM *p* = 7.88 × 10⁻⁴ (***p < 0.001). **e, f** FRAP analysis of condensates containing labeled ALIX and CaLB1. Photobleaching was performed either on ALIX (**e**) or CaLB1 (**f**). Arrowhead, bleached condensate. s: seconds. The experiment was conducted three times, and representative images are shown. **g, h** Quantification of the results in (**e, f**). The line shows the mean value of all measured condensates and error bands indicate standard deviation. On the fit mean data (R Square: 0.99), the τ₁/₂ values (half time of recovery) and the proportion of the mobile fraction were calculated. s: seconds. 6 and 8 condensates were analyzed for ALIX and CaLB1, respectively. **c, d, g, h** Source data are provided as a Source Data file. **c, d** Box plot: center line, median; box limits, first and third quartiles; whiskers, 1.5x interquartile range. Two-tailed *t*-test, no equal variance.

presence of CaLB1 did not affect the mobility of ALIX in condensates (Fig. 7e–h, Supplementary Fig. 7c). Altogether, these results suggest that CaLB1 and ALIX can undergo phase separation to form condensates both in vitro and in vivo and that the IDR of CaLB1 is necessary for the efficient assembly of CaLB1 and ALIX into condensates.

## CaLB1 and ALIX modulate maturation of autophagosomes

To test if CaLB1 is consistently associated with autophagosomes and delivered to the vacuole, 5-day-old seedlings expressing CaLB1-GFP were grown for an additional two days on a medium supplemented with 100 mM NaCl, then incubated with the vacuolar protease inhibitor E-64d. Autophagic bodies in the vacuole were subsequently visualized with 50 μM monodansylcadaverine (MDC). Whereas MDC-stained autophagic bodies were visible in the vacuoles of root epidermis cells, CaLB1-GFP signals was not apparent on these structures (Fig. 8a). The MDC staining overlapped with GFP-ATG8a signals and was dependent on functional autophagy, as MDC signals were not observed in the autophagy-deficient *atg10-1* cells (Fig. 8f and Supplementary Fig. 8a, b).

To understand whether CaLB1 and ALIX are, like the cargo receptor NBR1, delivered to the vacuole upon salt-induced autophagy, we analyzed protein levels of CaLB1-GFP and GFP-ALIX upon salt treatment with or without E-64d that inhibit vacuolar proteases. Whereas NBR1 accumulated upon E-64d treatment, a similar

accumulation of CaLB1-GFP was not observed (Fig. 8b, c), indicating that CaLB1 is not efficiently targeted to the vacuole. For GFP-ALIX, no significant difference to NBR1 was observed for the three replicates (Fig. 8c).

We then investigated whether CaLB1 plays a role in the maturation and delivery of autophagic bodies upon salt treatment. We first examined the number of autophagosomes in wild-type and *calb1-1* seedlings treated with 150 mM NaCl for 2 h. *calb1-1* showed a higher increase in the number of autophagosomes upon salt treatment (Fig. 8d, e and Supplementary Fig. 8c), suggesting that *calb1-1* mutants may have defects in the delivery of salt stress-induced autophagosomes to the vacuole.

To test this hypothesis, we treated wild type, *calb1-1*, and *calb1-1* complemented with *CaLB1pro: CaLB1-GFP* with 100 mM NaCl for 2 days, incubated them with 100 μM E-64d and visualized autophagic bodies with MDC. The size of autophagic bodies in wild-type and CaLB1-GFP-expressing *calb1-1* was similar, whereas a significant decrease in the area of MDC-stained autophagic bodies was observed in *calb1-1* (Fig. 8f, g). This result indicates that the function of CaLB1 is required for proper delivery of autophagosomes in salt-induced autophagy.

The accumulation of ATG8a-positive puncta in *calb1-1* could be due to increased autophagy induction. We therefore monitored the expression of all nine *ATG8* genes in wild-type and *calb1-1* seedlings

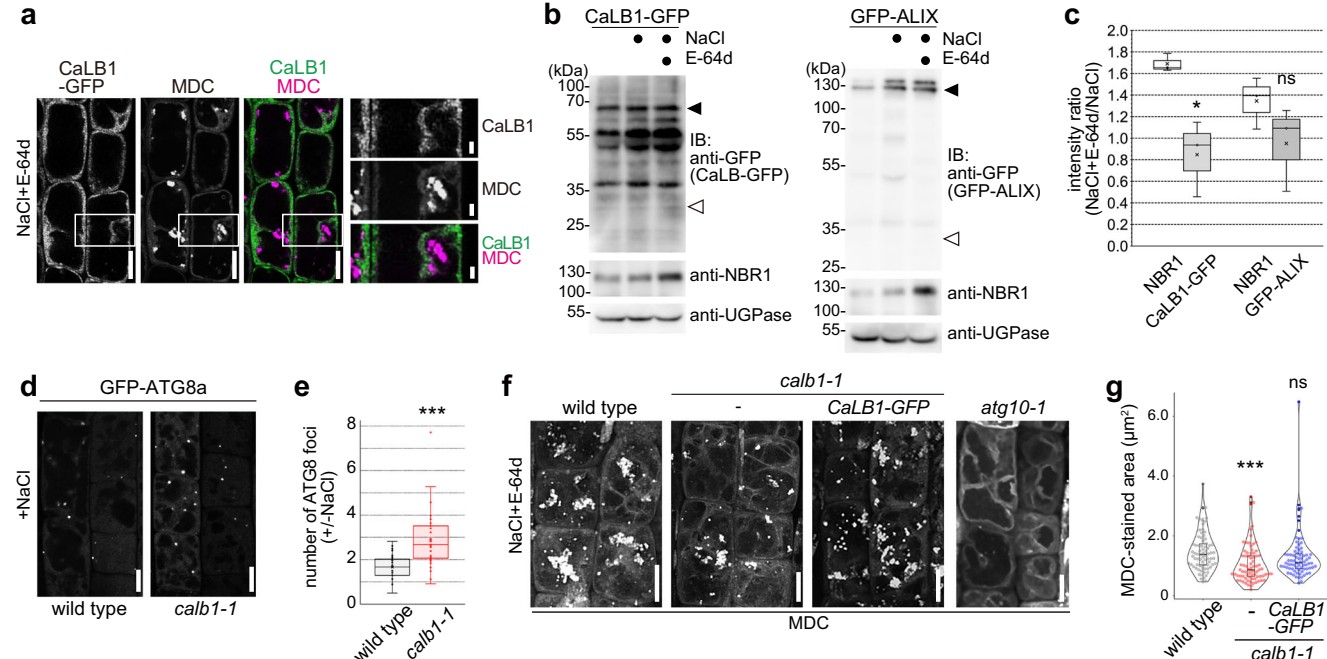

**Fig. 8 | CaLB1 modulates autophagosome maturation. a** *calb1-1* seedlings with *CaLB1pro: CaLB1-GFP* were treated for 2 days with 100 mM NaCl before the addition of 100 μM E-64d and autophagic bodies were stained with MDC. Rectangle: magnified region. The experiment was conducted twice, and one representative image is shown. Scale bars: 10 μm and 2 μm (magnification). **b** Seedlings expressing *CaLB1pro: CaLB1-GFP* and *ALIXpro: GFP-ALIX* were treated with either 150 mM NaCl or 150 mM NaCl and 100 μM E-64d for 2 h. Total extracts were subjected to immunoblotting with anti-GFP and anti-NBR1 antibodies. UGPase was used as loading control. The experiment was conducted three times, and one representative result is shown. Filled arrowhead: position of CaLB1-GFP or ALIX-GFP, open arrowhead: position of GFP. **c** The ratio of the intensities [NaCl+E-64d/NaCl] was calculated on three immunoblots conducted as in (**b**) and all values are shown. NBR1/CaLB1-GFP *p* = 0.0474 (*0.01*p* < 0.05), NBR1/GFP-ALIX *p* = 0.228 (ns: not significant, *p* > 0.05). **d** Seedlings expressing *UBQ10pro: GFP-ATG8a* were incubated with 150 mM NaCl for 2 h and root epidermis cells were analyzed using a confocal microscope. The experiment was conducted three times, and one representative

image is shown. Scale bars: 10 μm. **e** The number of GFP-ATG8a-positive foci +/−NaCl in (**d**) is shown in a box plot. *n* = 35 NaCl-treated roots were analyzed. The number of autophagosomes after salt treatment was divided by the average of the number of autophagosomes in the untreated roots. Wild type/*calb1-1* *p* = 0.00000479 (****p* < 0.001). **f** Wild-type, *calb1-1*, *calb1-1* with *CaLB1pro: CaLB1-GFP*, and *atg10-1* seedlings were treated with 100 mM NaCl for 2 days and subsequently with 100 μM E-64d. Autophagic bodies were stained with MDC and root epidermis cells were imaged using a confocal microscope. Maximal projection images are shown. The experiment was conducted twice, and one representative image is shown. Scale bars: 10 μm. **g** Quantification of the results shown in (**f**). *atg10-1* was omitted from the analysis as there was no accumulation of autophagic bodies. For each genotype, 15 seedlings (5 cells per seedling, *n* = 75) were analyzed. wild type/*calb1-1* *p* = 0.000132 (****p* < 0.001), wild type/*calb1-1* with *CaLB1pro:CaLB1-GFP* *p* = 0.199 (ns: not significant). **b**, **c**, **e**, **g** Source data are provided as a Source Data file. **c**, **e**, **g** Box plot: center line, median; box limits, first and third quartiles; whiskers, 1.5x interquartile range. Two-tailed *t*-test, no equal variance.

upon treatment with 150 mM NaCl for 90 min. Quantitative real time (qRT)-PCR analyses showed that the expression of all *ATG8s* was not significantly increased in *calb1-1* when compared to the wild type, except for *ATG8g* that was reduced in *calb1-1* (Supplementary Fig. 8d), suggesting that increased autophagy induction is not the cause for autophagosome accumulation.

To investigate the requirement of CaLB1 in the maturation of autophagosomes, we analyzed ESCRT-III localization in wild type and *calb1-1*. For this purpose, we generated a construct expressing a GFP- or mRFP-tagged VPS2.1 with a long linker sequence containing a 3xFLAG tag to avoid steric hindrance of the tag during VPS2.1 polymerization and filament formation[50] and tested whether it could complement the *ups2.1* null mutant which arrest growth during embryogenesis[51]. We genotyped the progeny of a VPS2.1/*vps2.1* plant carrying *VPS2.1pro: VPS2.1-3xFLAG-GFP* and found that homozygous *ups2.1* mutant seedlings carrying VPS2.1-*3xFLAG*-GFP could be recovered and that they were indistinguishable from the wild type (Supplementary Fig. 8e–g), showing that the generated *VPS2.1pro: VPS2.1-3xFLAG-GFP* construct is functional.

Next, we generated plant lines co-expressing *UBQ10pro: GFP-ATG8a* and *VPS2.1pro: VPS2.1-3xFLAG-mRFP* in wild type and *calb1-1*. After treating the seedlings with 150 mM NaCl for 2 h, VPS2.1-mRFP signals were observed on ATG8a-marked autophagosomes (Fig. 9a). However, the number of events was limited a small proportion of observed autophagosomes, most probably due to its transient nature. With 3.9% of autophagosomes (*n* = 2351) in the wild type and 2.3% of

autophagosomes (*n* = 1660) in *calb1-1* showed ESCRT signals (*p* = 0.067 in a two-tailored *t*-test), a significant difference could not be found between the two genotypes.

We therefore analyzed autophagosome closure at the ultra-structure level in wild-type or *calb1-1* seedling roots using transmission electron microscopy. We observed that in the *calb1-1* seedlings 62.9% of autophagosomes analyzed were open and 37.1% were closed (*n* = 70 autophagosomes). The number of open autophagosomes was significantly higher than in the wild type where we observed 42.6% of open autophagosomes and 57.4% of closed autophagosomes (*n* = 68 autophagosomes) (Fig. 9b and Supplementary Fig. 8h). Although we observed a significant difference between wild-type and *calb1-1*, we are aware that the analysis of three-dimensional autophagosomes by two-dimensional ultrathin sections could lead to biased results[3,4].

Thus, to further verify the role of CaLB1 in autophagosome maturation, we performed a protease-protection assay. Autophagosome-enriched fractions were prepared from total proteins from wild type and *calb1-1* treated with 150 mM NaCl for 2 h and treated with Proteinase K. The impaired closure of autophagosomes should result in increased degradation of unprotected GFP-ATG8a upon protease treatment, whereas GFP-ATG8 sequestered within the mature autophagosomes remains protected[3,38]. Simultaneous treatment with the detergent Triton X-100 disrupts the membrane. Although the level of GFP-ATG8a was comparable in wild type and *calb1-1*, the recovery of membrane-bound GFP-ATG8a was less efficient in *calb1-1* compared with the wild type (Fig. 9c). Upon Proteinase K

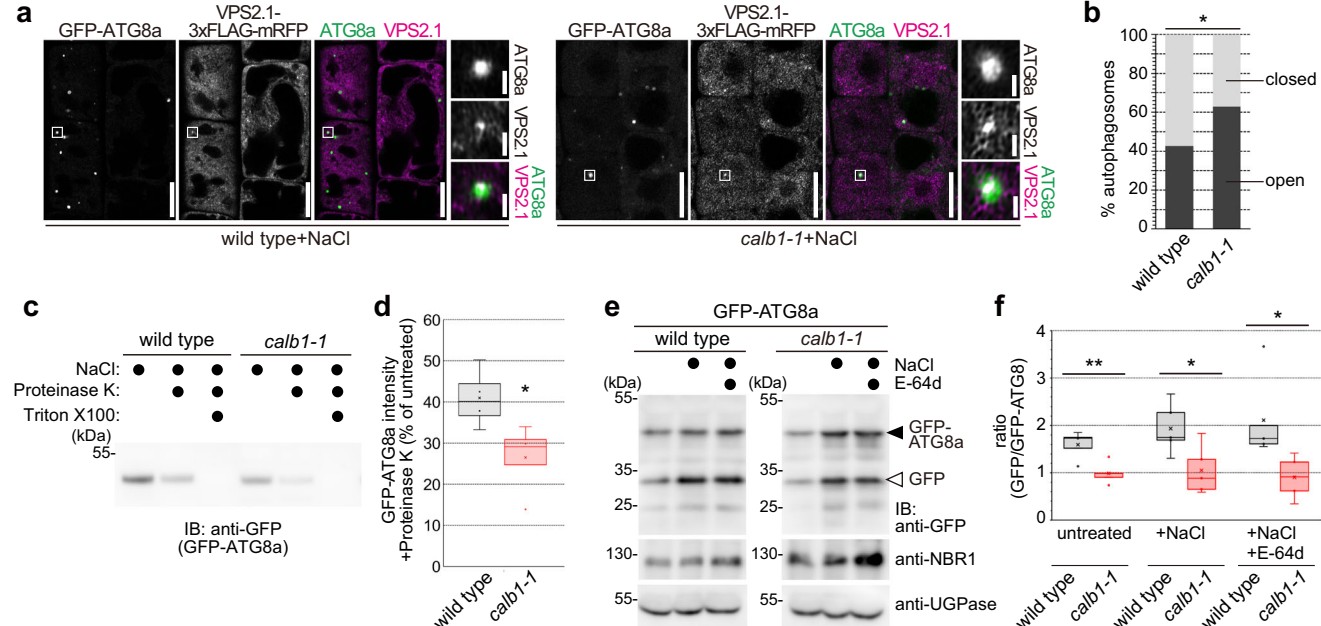

**Fig. 9 | CaLB1 modulates autophagosome maturation. a** Colocalization of ESCRT-III with autophagosomes. 5-day-old wild-type and *calb1-1* seedlings expressing *UBQ10pro:GFP-ATG8a* and *VPS2.1pro:VPS2.1-3xFLAG-mRFP* were incubated in 1/2 MS media containing 150 mM NaCl for two hours. Root epidermis cells were analyzed using a confocal microscope. Rectangle: magnified region. Scale bars: 10 μm and 1 μm (magnification). **b** Quantification of open and closed autophagosomes at the ultrastructure level in wild-type or *calb1-1* seedling roots using transmission electron microscopy. 5-day-old seedlings expressing were treated for 2 h in 1/2 MS supplemented with 150 mM NaCl before cryofixation. The analysis was conducted on sections taken from 4 different roots (wild-type, *n* = 68 autophagosomes; *calb1-1*, *n* = 70 autophagosomes). Wild type/*calb1-1 p* = 0.0174 (*0.01 < *p* < 0.05), two-tailed two proportion z-test. **c** Autophagosomes were isolated from wild-type and *calb1-1* seedlings and treated with Proteinase K or Proteinase K and Triton X-100 and subjected to immunoblotting using and anti-GFP antibody. The experiment was conducted four times, and one representative result

is shown. **d** Box plot shows the quantification of the results in (**c**). The intensity of the non-proteinase K sample was set at 100%. 4 replicates were analyzed, and all results are shown. Wild type/*calb1-1* (+Proteinase K) *p* = 0.0453 (*0.01 < *p* < 0.05). **e** GFP cleavage assay. 7-day-old seedlings were incubated in 1/2 MS, or 1/2 MS supplemented with either 150 mM NaCl or 150 mM NaCl and 100 μM E-64d for two hours. GFP-ATG8a and free GFP in total extracts of wild type and *calb1-1* were detected by immunoblotting along with NBR1. UGPase was used as loading control. The experiment was conducted five times, and one representative result is shown. **f** Box plot shows the quantification of the results in (**e**). Five replicates were analyzed, and all results are shown. wild type/*calb1-1* (untreated) *p* = 0.00610 (**0.001 < *p* < 0.01), wild type/*calb1-1* (+NaCl) *p* = 0.0282 (*0.01 < *p* < 0.05), wild type/*calb1-1* (+NaCl+E-64d) *p* = 0.0352 (*0.01 < *p* < 0.05). **b**, **c**, **d**, **e**, **f** Source data are provided as a Source Data file. **d**, **f** Box plot: center line, median; box limits, first and third quartiles; whiskers, 1.5x interquartile range. Two-tailed *t*-test, no equal variance.

treatment of the autophagosome-enriched fraction of *calb1-1*, we observed an increased degradation of GFP-ATG8a compared with the wild type (Fig. 9c, d), indicating more unclosed autophagosomes in *calb1-1* and suggesting that autophagosome maturation is impaired in *calb1-1*.

We also monitored the autophagic flux in *calb1-1*. For this end, the ratio of GFP-ATG8 and free GFP was examined in wild type and *calb1-1* seedlings (Fig. 9e). Quantification of the results showed that *calb1-1* had a significantly lower turnover of GFP-ATG8 compared with the wild type in all tested conditions (Fig. 9f).

Collectively, the data show that *CaLB1* is induced upon salt stress, and that CaLB1 interacts and localizes with ALIX on autophagosomes and undergoes phase separation with ALIX. Our data also show that CaLB1 can trigger the efficient assembly of ALIX into molecular condensates. The accumulation of ALIX into condensates could facilitate or increase the interaction between ALIX and ESCRT components and thus the crowding of the ESCRTs at the site of autophagosome closure (Fig. 10).

## Discussion

Autophagosome formation and biogenesis has been intensively studied[52,53], and although it is widely accepted that the membrane scission activity of ESCRTs is important for autophagy, the exact molecular basis for their role in the maturation of autophagosomes has not yet been well understood[11]. In mammals, the ESCRT-III subunit CHMP2A was shown to be recruited to closing autophagosomes by the ESCRT-I subunit VPS37A[6]. In yeast, the ESCRT-III subunit Snf7 was reported to be recruited to autophagosomal membranes by interacting with the fungi-specific protein scaffold protein Atg17[3]. In *Arabidopsis*, the plant-specific ESCRT component FYVE1/FYVE DOMAIN PROTEIN REQUIRED FOR ENDOSOMAL SORTIN 1 (FREE1) was shown to

directly interact with ATG8 and ESCRT-I and was proposed to mediate the ESCRT-III recruitment to autophagosomes[7].

On many cellular compartments where membrane remodeling takes place, multiple ESCRT-interacting proteins are known to work together to mediate the efficient recruitment of ESCRT-III. For example, to ensure correct and timely membrane scission during endosomal cargo sorting and cytokinetic abscission in yeast and human cell lines, ALIX plays an important role in the recruitment of ESCRT-III proteins in parallel with ESCRT-I and ESCRT-II[54,55]. The activity of ESCRT-III at the plasma membrane or damaged lysosomes requires the synergistic function of both ALIX and ESCRT-I. Similarly, CaLB1 and ALIX, together with other ESCRT components, could provide a FREE1-independent mechanism to modulate ESCRT-III function at autophagosomes specifically under salt stress condition or could coordinate the ESCRT-dependent maturation of autophagosomes together with FREE1. As both FREE1 and ALIX are involved in a wide range of membrane transport processes, it will be challenging to dissect these possibilities using currently available mutant lines.

By binding to the autophagosomal membrane through interactions with ATG8s and PI(3)P and by stimulating the condensate formation of ALIX, CaLB1 could facilitate the recruitment of ESCRTs to the closing autophagosomes. Phase transition was shown to play a role in membrane reformation events involving ESCRTs at the endosomal and nuclear membranes. On endosomal membranes, ESCRT-III recruitment and polymerization were shown to be driven by the condensation of ESCRT-0 and polyubiquitin[56]. At the nuclear membrane, the low complexity region of an inner nuclear membrane protein LEM2 was shown to trigger its phase separation[57]. This leads to the oligomerization of the LEM2-interacting ESCRT protein CHMP7 into macromolecular rings, which in turn serve as the basis for the ESCRT-III filaments for membrane scission and repair[57].

In a recent publication, human ALIX was shown to undergo phase separation with ESCRT-III subunits and transition rapidly to a non-dynamic phase, which was suggested to be important for the recruitment and subsequent assembly of ESCRT filaments[46]. Arabidopsis ALIX was largely immobile in condensates, as reported for its human ortholog. For human ALIX, hyperphosphorylation in the C-terminal disordered region was shown to regulate its assembly into condensates[46]. Several of the tyrosine residues in the C-terminus tail are also conserved in Arabidopsis ALIX. Given the absence of dedicated tyrosine kinases in plants, it is intriguing to understand whether plant ALIX undergoes a similar phosphorylation-based regulation.

As condensates can wet membranes and affect the membrane topology[47], the formation of CaLB1 and ALIX-containing condensates on the autophagic membranes could affect local membrane tension and topology to facilitate the membrane scission processes. A recent report showed that phase separation plays a role in orchestrating the ordered assembly of proteins at the plasma membrane during endocytosis in Arabidopsis[58]. Although yet little is known about the function of molecular condensates in intracellular trafficking in plants, accumulating evidence suggest that they represent an additional layer of regulation for biomolecules and are undoubtedly involved in a wide range of cellular processes.

The mechanisms of triggering the condensation of CaLB1 and ALIX and positioning CaLB1 and ALIX-containing droplets are still unknown. Notably, CaLB1 and ALIX were found in a defined punctum on the autophagosomal membrane. The localization of CaLB1 and ALIX must be defined by additional proteins, local lipid compositions, or the membrane topology, as ATG8s and PI(3)P that interact with CaLB1 are distributed along the phagophore. CaLB1-containing droplets were not stably associated with the autophagosomes. This suggests that CaLB1 and ALIX may only be recruited at a certain stage during autophagosome maturation. Alternatively, they may only be able to stably associate with membranes with specific topologies, such as the site of membrane closure. In yeast, the Atg24 complex was recently reported

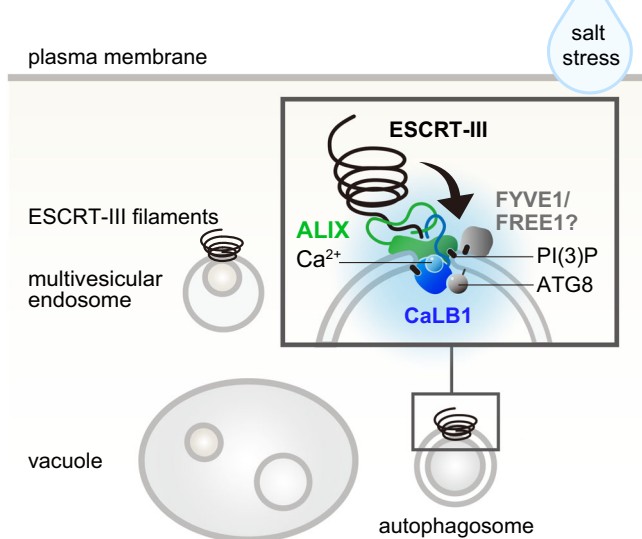

**Fig. 10 | Model for CaLB1 function in autophagy.** *CaLB1* is induced by salt and is necessary for resistance to salt stress. The C2 domain of CaLB1 binds the phospholipid PI(3)P and ATG8 proteins, which are present in autophagosomal membranes. This could form the basis of CaLB1 recruitment to autophagosomes. With the C-terminal IDR, CaLB1 binds the IDR of ALIX. CaLB1 can form molecular condensates on autophagosomes in conjunction with ALIX. The stimulation of condensate formation by CaLB1 may serve as a trigger for the formation of ALIX filaments, which in turn facilitates the subsequent recruitment and positioning of ESCRT-III filaments through multivalent interactions at the site of autophagosome closure. Together with other ESCRT-related proteins, CaLB1 and ALIX present important factors that coordinate autophagosome maturation in plant cells, especially under salt stress.

to localize at the edge of the phagophore and function as part of the rim aperture stabilizing the phagophore membrane[59,60]. It remains to be investigated whether and how the components of the rim aperture function together with CaLB1, ALIX and orchestrate the recruitment of the ESCRT machinery during autophagosome maturation in plants.

The expression of *CaLB1* is induced by NaCl treatment and mutants of *CaLB1* are more sensitive to NaCl compared with the wild type, indicating that CaLB1 function may be particularly important under salt stress. Human ALIX has been shown to mediate membrane repair and reformation under various stress conditions, such as in response to a $Ca^{2+}$ influx into the cytosol after membrane damage, which triggers the recruitment of ALG-2, and subsequently that of the ESCRT machinery, to the site of rupture[61–63]. The observed relocalization of CaLB1 to the plasma membrane upon salt treatment could indicate that CaLB1 function may not be limited to autophagosomes. Whether CaLB1 and ALIX functions together in other membrane-mediated processes other than autophagy in plants remains to be investigated.

## Methods

All primers used in this study are listed in Supplementary Table 1, and all plasmids used in this study are listed in Supplementary Table 2.

### Molecular cloning

The coding sequence (CDS) of the Arabidopsis genes used for cloning was obtained by mRNA extraction from seedlings using the NucleoSpin RNA Plant kit (Macherey-Nagel, 740949.50) followed by retrotranscription using M-MuLV reverse transcriptase [New England Biolabs (NEB)]. The resulting cDNA was used as the template for further PCR reactions. The genomic sequences of the cloned genes were obtained by PCR from genomic DNA isolated from Arabidopsis seedlings.

To generate the plasmids expressing *GAD-HA-CaLB1(FL)* (pKK90), *GAD-HA-CaLB1(C2)* (pNM39), *GAD-HA-CaLB1(PRD)* (pNM40), *GAD-HA-CaLB1(1-177)* (pNM41), and *GAD-HA-CaLB1(1-220)* (pNM42) for YTH analysis, fragments or the complete CDS of *CaLB1* were amplified, using the primers KK156/KK157, KK156/NM112, NM113/KK157, NM130/NM132, and NM130/NM133, respectively. The PCR products were cloned into *pGADT7 AD* (Takara Bio, 630442) with restriction enzymes NdeI and BamHI (FL, C2, PRD) or NdeI and EcoRI (1-177, 1-220). The plasmids *GBD-Myc-ALIX(PRD)* (pTB32) and *GBD-Myc-ATG8i* (pTB70) were generated by amplifying the CDS of *ALIX* and *ATG8i* with primers TB99/TB100 and TB175/TB176, respectively. The PCR products were cloned into *pGBKT7* (Takara Bio, 630443) using the restriction enzymes NdeI and BamHI (ALIX) or NdeI and SalI (ATG8i).

To generate *GAD-HA-CaLB1[Y34A]* (pYP1), *GAD-HA-CaLB1[F64A]* (pYP2), *GAD-HA-CaLB1[Y172A]* (pYP3), and *GAD-HA-CaLB1[Y205A]* (pYP4), mutations in *CaLB1* were introduced by a two-step Overlap PCR using TB245/TB246, TB247/TB248, TB249/TB250, and TB251/TB252, respectively. The CDS was subsequently amplified using TB101/TB102 and cloned into *pGADT7 AD* using restriction enzymes NdeI and BamHI.

For *CaLB1-6xHis* (pNM7) and *CaLB1(C2)−6xHis* (pNM45), the CDS of *CaLB1* and the C2 domain of *CaLB1* were amplified using primers KK156/NM27 and NM130/NM149, respectively, and cloned into *pET21a(+)* (Merck, 69740) between NdeI and HindIII sites. To generate the construct *CaLB1(codon optimized)−6xHis* (pNM66), the codon optimized sequence of *CaLB1* (Eurofins Genomics) (pNM64) was amplified with primers NM188/NM189 and cloned into *pET21a(+)* between the NdeI and HindIII sites.

For the generation of *CaLB1[D32A]−6xHis* (pJW6), the CDS of *CaLB1* was cut out from pNM7 with XbaI and HindIII and cloned between the XbaI and HindIII sites of *pUC19* (Takara Bio, 3219) to obtain pNM36. The D32A mutation was introduced using primers NM121/NM122. The D32A mutated CDS of *CaLB1* was cut out from

CaLB1[D32A] (pJW3) with XbaI and HindIII and cloned between the XbaI and HindIII sites of pET21a(+). For *CaLB1(C2)[D32A]−6xHis* (pNM101), the C2 domain of *CaLB1* was amplified with the primers NM130/NM149 using pJW6 as a template and cloned between the NdeI and HindIII sites of *pET21a(+)*. To generate *CaLB1(1-177)−6xHis* (pNM46), a fragment of the CDS of *CaLB1* was amplified using the primers NM130/NM150. The PCR fragment was cloned into *pET21a(+)* with restriction enzymes NdeI and HindIII.

For *CaLB1(1-177)[C49S]* (pNM93), mutation in *CaLB1* was introduced by two-step Overlap PCR using GP17/NM272, NM271/GP16 and pNM93 as a template. The CDS was subsequently amplified using NM130/NM150 and cloned into *pET21a(+)* using restriction enzymes NdeI and HindIII. The plasmid pNM93 was used for the generation of the construct *CaLB1(1-177)[C49S/C107S]−6xHis* (pNM97). Mutation on CaLB1 was inserted with the primers GP17/NM274 and NM273/GP16. The CDS was then generated using the primers NM130/NM150 and cloned between the NdeI and HindIII sites of *pET21a(+)*. To generate the constructs *CaLB1(C2)[C49S/C107S/S29C]−6xHis* (pNM125), *CaLB1(C2)[C49S/C107S/L83C]* (pNM127), and *CaLB1(C2)[C49S/C107S/N55C]−6xHis* (pNM129), mutations were introduced by overlap PCR using pNM97 as a template with the primers NM350/NM351, NM354/NM355, and NM342/NM343, respectively. The CDS of *CaLB1* was then amplified using GP17/NM149 and cloned into *pET21a(+)* using the restriction sites NdeI-HindIII.

To obtain pTB111, an EcoRI site was introduced into *pGEX-6P1* (Merck, GE28-9546-48) by site-directed mutagenesis using primers TB306F/TB306R to obtain pTB103. The multiple cloning site, *GST*, and the PreScission Protease recognition site were amplified by PCR with primers TB307/TB308, TB311/TB312, and TB309/TB310, respectively, and digested with the restriction enzymes EcoRI/BsaI, BsaI/BsaI and BsaI/XhoI, respectively. The fragments were then inserted into the EcoRI/XhoI sites of pTB103 to obtain pTB111. To produce *CaLB1-GST* (pNM68), the codon optimized CDS of *CaLB1* (Eurofins Genomics) (pNM64) was amplified with primers NM188/NM237 and cloned between the NdeI/SalI sites of pTB111.

For *GST-ALIX* (pKK48) and *GST-ScBro1* (pNM54) and *GST-HsALIX* (pNM48) constructs, the CDS of Arabidopsis *ALIX*, the genomic sequence of yeast *Bro1*, and the CDS of human *ALIX* were amplified by PCR using the primers KK40/KK41, NM163/NM164, and NM127/NM128, respectively. *ALIX* was cloned into *pGEX-6P1* using BamHI/SalI while *ScBro1* and *HsALIX* were cloned into *pGEX-6P1* (Cytiva) using BamHI/EcoRI. For *MBP-ALIX(ΔBRO1)* (pKK32), the CDS of Arabidopsis *ALIX* was amplified with the primers KK89/KK41 and cloned into *pMAL-p2p* using the restriction enzymes BamHI and SalI. *GST-ALIX* (codon optimized) (pNM67) was generated by the amplification of the codon-optimized CDS of *ALIX* (Eurofins Genomics−pNM65) using the primers NM190/NM191 followed by the cloning into *pGEX-6P1* between the restriction sites BamHI/EcoRI.

To generate *6xHis-T7tag-ATG8a* (pNL7), *6xHis-T7tag-ATG8f* (pNL8), and *6xHis-T7tag-ATG8i* (pLK3), the CDS of *ATG8s* was amplified by PCR using the primers TB161/NL47, TB169/NL49, and TB175/TB176, respectively, and cloned into *pET28a(+)* (Merck, 69864) using the restriction enzymes NdeI and SalI.

To generate the CRISPR construct *U6-26pro:sgRNA(At4g34150_exon1)-U6-26term* (pNM10), the target sequence TATACAAGGCCAGATTCTCGAGG was selected using the web tool CHOPCHOP (https://chopchop.cbu.uib.no)[64]. The target sequence was inserted by annealing the oligonucleotides NM46/NM47 and cloning the annealed fragments into *pHEE401E*[65] by a Golden Gate reaction.

For the generation of *CaLB1pro:g(genomic)CaLB1-eGFP* (pNM22), the genomic sequence of *CaLB1* including the region starting from 1007 bp upstream of the start codon, was amplified with primers NM78/NM80 and cloned into *pENTR™/D-TOPO™* using the pENTR™/D-TOPO™ cloning kit (Thermo Fisher, K240020). The resulting plasmid pNM17 was used together with the destination vector *pFAST-R07*[66] for

Gateway™ cloning (Thermo Fisher, Gateway BP Clonase II Enzyme mix, 11789020, Gateway BP Clonase II Enzyme mix, 11791020). To generate *CaLB1pro:gCaLB1[F64A]-eGFP* (pNM117), the genomic sequence of *CaLB1* was subjected to mutagenesis by overlapping PCR using primers NM31/TB248 and TB247/NM80 followed by a PCR reaction using primers NM31/NM80. The resulting PCR product was cloned between the restriction sites XhoI/BstXI of pNM22.

*CaLB1pro:gCaLB1-mRFP* (pNM52) was generated by a Golden Gate reaction. To remove a BsaI site from the promoter region, overlapping PCR was performed using primes NM89/NM92 and NM91/NM90. For mutating a BsaI site in the genomic sequence of *CaLB1*, overlapping PCR was performed using primers NM93/NM158 and NM157/NM94. The *mRFP: T3SS* sequence was obtained by PCR with the primers NM159/NM160 from *pUBC-RFP*[67]. *NOSpro: Basta*[R] was amplified from *pGWB614*[68] using the primers NM99/NM100. The fragments were subsequently ligated into *LIIα F1-2*[69].

For the generation of *UBQ10pro:mRFP-ATG8a* (pNM106), *UBQ10pro:eGFP-ATG8a* (pNM107), *UBQ10pro:mRFP-ATG8e* (pNM108), and *UBQ10pro:mRFP-ATG8i* (pNM110), the coding sequence of the *ATG8*s was amplified with primers NM295/NM296, NM297/NM298, and NM299/NM300, respectively. The obtained PCR products were cloned into *pDONR™207* (Thermo Fischer Scientific) to obtain entry clones pNM103, pNM104, and pNM105, respectively, which were then used for an LR-reaction with either *pUBN-RFP* or *pUBN-GFP*[67].

*VPS2.1pro: VPS2.1-3xFLAG-GFP* (pFG41) and *VPS2.1pro: VPS2.1-3xFLAG-mRFP* (pFG46) were generated by a Golden Gate cloning. The promoter and the genomic sequence of *VPS2.1* were amplified with primers MN522/MN523 and MN528/MN529, respectively, and the PCR products were directly used for a Golden Gate reaction. The *3xFLAG* tag was inserted into *pJET1.2* (Thermo Fisher Scientific, K1231) after amplification of the sequence with the primers with the primers TB179/TB180 to obtain pTB66. The *3xFLAG-GFP* (pFG44) and *3xFLAG-mRFP* (pFG45) units were generated by using pTB66 and *pUBC-GFP-Dest*[67] or pTB66 and *3xFLAG* and *pUBC-mRFP-Dest*[67] as templates with primers FG87/FG88 and FG89/FG90, or TB179/TB180 and FG87/FG106, respectively. For pFG45, *3xFLAG* and *mRFP* were combined by overlapping PCR using the primers FG105/FG107 and cloned into *pJET1.2*. The *NOS* terminator was amplified from the vector *pFAST-R07*[66] with primers TB141/TB142 and cloned into *pJet1.2* (pTB63). The selection marker *OLE1Pro: OLE1-TagRFP* with a *NOS* terminator downstream of its sequence was amplified from the vector *pFAST-R07*[66]. For mutating BsaI sites in the promoter, in the *OLE1* sequence, and the *TagRFP* sequence, overlapping PCR was performed using primers TB277/TB278, TB275/TB276, and TB273/TB274, respectively. The whole reporter cassette was amplified with the primers TB270 and TB271 and subcloned into *pJET1.2* (pTB85). The plasmids pFG44 or pFG45, pTB63, pTB85 and the genomic sequence of *VPS2.1* were used together with the vector *pBB10*[69] for Golden Gate assembly.

CRISPR constructs for protoplasts, CRISPR(CaLB1) (pMN201) and CRISPR(mutCaLB) (pMN202), were generated by Golden Gate reaction. *U6-26* promoter and terminator were amplified either from the plasmid pNM10 together with the CRISPR target sequence ATACAAGGCCAGATTCTCG using the primers MN439/MN440 or generated by overlap PCR together with the mutated target sequence ATTCAAGGTCCGCTTATCG with primers MN572 and MN573. The PCR products were subcloned into *pJET1.2* obtaining pMN199 and pMN200, respectively. The *35S* promoter was amplified from *pGWB411*[70] with the primers MN443/MN444 and subcloned into *pJET1.2* to generate pMN173. The *CAS9* coding sequence was amplified from the vector *pHEE401E*[65] with the primers MN445 and MN446 cloned into *pJET1.2* obtaining pMN177. The *NOS* terminator was amplified from the vector *pGWB411* with primers MN447/MN448 and cloned into *pJET1.2* to generate pMN174. The constructs pMN173, pMN174, and pMN175 together with the vector *pBBO2*[71] were used for Golden Gate assembly with either pMN199 or pMN200.

For the cloning of *35Spro: GFP-ALIX* (pTB204), the CDS of *ALIX* was amplified from Arabidopsis cDNA with the primers NM196 and TB450, containing overhangs for Golden Gate assembly. The sequence of *35S* promoter was amplified from *pFASTR-05*[69] with primers TB112/TB113 and the *sGFP* gene was amplified from *pGWB404*[68] with primers TB10/TB19. All the generated PCR products were used for the golden gate reaction into the vector *pBBO2* (*pUC57-gentamycin*[R]).

For the generation of *UBQ10pro: GFP-ALIX* (pMN224), the *UBQ10* promoter was amplified using the primers TB295/TB296 and cloned into *pJET1.2* to generate pTB90. The CDS of *ALIX* was amplified with the primers NM196/TB450 and cloned into *pJET1.2* to obtain pTB203. pTB187 was generated by amplifying the *sGFP* with Primers TB10/TB19. pTB90, pTB187, pTB203, and pTB63 were used together with the vector *pBBO2* for the golden gate assembly.

## Plant material and growth assays

All Arabidopsis lines used in this study are listed in the supplemental table and are in the Col-0 background. The marker lines *SYP43-pro:mRFP-SYP43*[72], *ARA6pro:ARA6-mRFP*[73], *ALIXpro:GFP-ALIX*[13], and *UBQ10Pro:GFP-ATG8a*[74] as well as the mutants *atg10-1*[75], *atg7-2*[76], *alix-2*[14], and *alix-4*[14] were established previously.

Arabidopsis seeds were surface sterilized in a solution of 1% NaOCl, stratified at 4 °C in dark for 1 to 3 days, and grown under long day (16 h light / 8 h dark), short day (8 h light / 16 h dark), or continuous light conditions at 21 °C in 1/2 MS [2.15 g/L Murashige and Skoog medium including vitamin B5 (Duchefa, M0231), 0.5 g/L MES, pH 5.7].

To generate *calb1-1 and calb1-2*, wild-type plants were transformed pNM10 by Agrobacterium-mediated floral-dip method[77]. Homozygous mutant*s* were identified using high-resolution melting analysis and confirmed by nucleotide sequencing of the region amplified by primers NM31/NM32 or alternatively digested with restriction enzymes PspXI or XhoI. The T-DNA insertion line of *VPS2.1* was obtained from the GABI-Kat collection (GABI_670D06). For genotyping, PCRs were performed using primers FG156/FG155 and FG155/o8474. For identifying the complemented lines, primers FG156/FG154 were used. The presence of the *VPS2.1pro: VPS2.1-3xFLAG-GFP* was verified using primers FG79/GP41.

For the root length assay, Arabidopsis seedlings were grown for 7 days in long day on 1/2 MS after stratification for 3 days. Roots were measured using Fiji (https://imagej.net/software/fiji/)[78] on scanned images. The length of the primary roots was measured with the free-hand tool in Fiji[78] and graphs were prepared using Excel (Microsoft) or RStudio (http://www.rstudio.com/). For the carbon starvation (-C) experiments, 7-day-old seedlings grown in 1/2 MS were transferred to the dark for a period of 6 days at 21 °C. For treatment with NaCl, 1/2 MS media were supplemented with the indicated concentration of NaCl and seedlings were incubated for the indicated time for each experiment.

## Chlorophyll content measurement

For chlorophyll content measurement, seedlings were incubated in N,N-dimethylformamide at 4 °C with agitation in the dark for 48 h. 150 μL of the supernatant was used to measure the absorption at 664 nm and 647 nm, and the total chlorophyll content was calculated as follows: total chlorophyll (μg/g$^{-1}$ fresh weight) = [(OD664 × 7.04) + (OD647 × 20.27)]/fresh weight.

## Yeast two-hybrid (YTH) assay

YTH assays were conducted using the yeast strain Y8800 (*MATa leu2-3 112 trp1-901 his3-200 ura3-S2 gal4Δ gal80Δ GAL2-ADE2 LYS2::GAL1-HIS3 MET2::GAL7-lacZ cyh2R*)[79]. Yeast transformants were selected on a synthetic dropout medium lacking leucine and tryptophan. Three independent colonies were grown in liquid medium and the cell density was adjusted to OD$_{600}$ = 1 before spotting. The auxotrophic growth was tested on synthetic dropout medium lacking leucine,

tryptophan, and histidine with or without 1 mM 3-amino-1,2,4-triazole as indicated.

## Recombinant protein purification and in vitro pull-down assay

Recombinant proteins were expressed in and purified from the *E.coli* Rosetta (DE3) strain (Merck, 70954) [*F-ompT hsdS$_B$*(r$_B^-$ m$_B^-$) *gal dcm (DE3) pRARE (Cam$^R$)*]. GST-fused proteins were purified using Glutathione Magnetic Agarose Beads (Thermo Fischer, 15585899) or Protino™ Glutathione Agarose 4B (Macherey-Nagel, 745500), His-fused proteins were isolated using TALON™ (TakaraBio, 635501) beads, and MBP-fused proteins were isolated using Amylose resin beads (NEB, E8021S).

For the pull-down assay using fragments of ALIX against CaLB1, TALON™ beads loaded with CaLB1-6xHis were incubated with 20 pmol of MBP or MBP-fused proteins in cold buffer A′ (50 mM Tris, 100 mM NaCl, 10%(w/v) glycerol, 0.1%(v/v) Triton X-100, pH7.5) under rotation at 4 °C. Beads were washed extensively with buffer A′ and bead-retained proteins were eluted in cold buffer supplemented with 400 mM of Imidazole (Merck, 104716). For analyzing in vitro binding of GST-ALIX and CaLB1-6xHis or CaLB1(C2)−6xHis, Protino™ Glutathione Agarose 4B were loaded with GST-ALIX and subsequently incubated with 75 pmol of His-tagged proteins for 2 h at 4 °C in buffer A (50 mM Tris, 150 mM NaCl, 10%(w/v) glycerol, 0.1%(v/v) Triton X-100, pH7.5) under rotation. After extensive washing, proteins were eluted with 50 mM glutathione. For in vitro binding assays with CaLB1-GST and His-ATG8s, TALON™ beads loaded with 6xHis-ATG8s were incubated with 100 pmol of either CaLB1-GST, GST-ALIX, or GST at 4 °C for 1 h in Talon Buffer (50 mM Na$_2$HPO$_4$, 300 mM NaCl,10% glycerol, pH7.0). After extensive washing, proteins were eluted with 150 mM Imidazole. Eluted samples were subjected to SDS-PAGE and analyzed by immunoblotting.

## Protein extraction, GFP-cleavage assays, and autophagosome enrichment

For protein detection after salt treatment and GFP-cleavage assays, total extracts were prepared from 10-day-old seedlings ground using a Tissuelyser II (QIAGEN, 85300). Samples were resuspended in buffer A supplemented with 0.2% (v/v) Triton X-100, 1 mM Phenylmethylsulfonylfluorid (PMSF), and Complete Protease Inhibitor Cocktail (Merck, 11836145001) and filtered through a polyamide filtration sheet with a pore size of 50 μm. The samples were then centrifuged at 500 x *g* at 4 °C for 15 min and subjected to immunoblotting.

For autophagosome enrichment and protease protection-assay, 1 to 5 g of 5 to 10-day-old seedlings were treated for two hours in liquid 1/2 MS supplemented with 150 mM NaCl and ground in cold GTEN-based buffer [10% glycerol, 30 mM Tris (pH 7.5), 1 mM EDTA, pH 8, 150 mM NaCl, 0.4 M sorbitol, 5 mM MgCl$_2$, 1 mM Dithiothreitol (DTT), protease inhibitor cocktail, and 1% Polyvinylpolypyrrolidon (PVPP)] with a ratio of 4:1 (v/w). Autophagosomes were enriched by centrifugation[38]. For the protease-protection assay, samples were centrifuged at 2000 x *g* at 4 °C for 10 min, and the supernatant was further centrifuged at 100,000 x *g* at 4 °C for 60 min and dissolved in a GTEN-based buffer without PVPP. Autophagosome-enriched fractions were treated with either Proteinase K (60 ng/μL) or with Proteinase K and 1% (v/v) Triton for 30 min on ice. The reaction was blocked by using 2.5 mM PMSF and cOmplete Protease Inhibitor Cocktail. Proteins were subsequently precipitated by adding Trichloroacetic acid and after centrifugation and washing with acetone, samples were subjected to immunoblotting.

## Immunoblotting and protein detection

Immunoblotting was conducted using the following primary antibodies: anti-ALIX[14], anti-H$^+$-ATPase (5000× diluted, Agrisera, AS07260), and anti-FLAG (M2) (1000×diluted, Merck, F1804), anti-GBD (3000× diluted, Santa Cruz, sc-510), anti-GFP (1000× diluted,

3H9, Chromotek, 3H9-100), anti-RFP (1000× diluted, 6G6, Chromotek, 6G6-150), anti-GST (1000× diluted, Eurogentec)[71], anti-His (1000× diluted, Thermo Fisher, P-21315), anti-HA(3F10) (1000× diluted, Roche, 11867423001), anti-MBP (10000× diluted, NEB, E8032S), anti-UGPase (3000× diluted, Agrisera, AS05086), anti-NBR1 (2000× diluted, Agrisera, AS142805A), anti-H3 (5000× diluted, Agrisera, AS10710), anti-ACTIN (50× diluted, JLA20, Merck, MABT219), anti-CDC2 (5000× diluted, Santa Cruz, sc-166885). As secondary antibodies, anti-rat-HRP (80000× diluted, Merck, A9037), anti-mouse-HRP (80000×diluted, Merck, A9044), anti-rabbit-HRP (80000×diluted, Merck, A0545), anti-rabbit-Alkaline Phosphatase (AP) (30000× diluted, Merck, A3812), and an anti-rat-AP (30000× diluted, Merck, A6066) were used and to detect protein bands in an Amersham™ Imager 600 (Cytiva). Uncropped and unprocessed pictures of the gels and immunoblot membranes are shown in the Source Data File.

## Microscale thermophoresis

For Microscale thermophoresis, Monolith NT.115 (NanoTemper) equipped with BLUE and GREEN detectors or Monolith X (NanoTemper) equipped with spectral shift and MST technologies with a RED detector were used. For ion binding experiments, recombinant CaLB1(C2)−6xHis and CaLB1(C2)[D32A]−6xHis in MST buffer (50 mM Tris pH 7.5, 150 mM NaCl, 0.05% Tween-20) were labeled with the Protein Labeling Kit GREEN-MALEIMIDE (NanoTemper, MO-L005) following the manufacturer instructions. Microscale Thermophoresis was measured with Monolith NT.115 using 50 nM CaLB1(C2) in the presence of increasing concentrations of either Ca$^{2+}$ or Mg$^{2+}$. For the analysis of CaLB1 with either ATG8i or ubiquitin, recombinant CaLB1(C2) was labeled with a NHS-RED protein labeling kit (NanoTemper, MO-L001 or MO-L011). 20 nM of labeled CaLB1(C2) was mixed with ATG8i or UB at a sequential dilution starting from 109 μM and 58.5 μM, respectively.

## Lipid overlay assay

Lipid overlay assays were performed using PIP Strips™ (Thermo Fisher, 10728234). Membranes were incubated for 4 to 6 h at 4 °C in a blocking solution (20 mM Tris pH7.4, 150 mM NaCl, 0.1% Tween, 3% fatty acid free bovine serum albumin (Roth, 0052) supplemented with 10 μM CaCl$_2$ or 5 mM EGTA as indicated. Subsequently, membranes were incubated with 1 μg/μL 6xHis- or GST-fused proteins overnight at 4 °C. The lipid-bound proteins were detected using an anti-His antibody or an anti-GST antibody.

## Microscopy and image analyses

Confocal microscopy of the fluorophore-fusion proteins and fluorescent dyes was conducted with LSM700 or with LSM880 equipped with AiryscanFast (Zeiss), a GaAsP PMT-Array detector, and a 63x/1.40 PlanApochromat (oil) objective. For the excitation of GFP and RFP, the 488 nm Argon ion laser and the 561 nm diode-pumped solid-state laser were used, respectively. Monodansylcadaverine (MDC, Merck, D4008) was excited at 405 nm, Alexa-488™ (Thermo Fisher, A20000) was excited at 488 nm, NT-GREEN (NanoTemper, Protein Labeling Kit GREEN-MALEIMIDE, MO-L005) was excited at 561 nm, and NT-RED (NanoTemper, NHS-RED protein labeling kit, MO-L001), was excited at 633 nm. Time series were obtained using the "time series" function in ZEN Black (Zeiss) with 2 s intervals. For quantification, a median filter with a radius of 1 pixel was applied on the 8-bit time series and a threshold filter was applied using the Otsu method. Images were obtained with the ZEN black software, processed with Photoshop (Adobe) and analyzed with Fiji[78].

For the colocalization analyses with the early and late endosomal markers, seedlings were treated either with 50 μM Brefeldin A (BFA) (Merck, B6542) for 1 h or with 33 μM Wortmannin (WM) (Applichem, W1628) for 2 h, respectively and dimethyl sulfoxide was used as a control.

For salt treatment, 5-day-old seedlings were transferred from 1/2 MS plates to liquid 1/2 MS either with or without 150 mM of NaCl for 2 h before imaging. For counting the number of CaLB1 or ALIX foci or the number of autophagosomes, a threshold was set for each image according to the signal strength. The images were then converted to binary images, masked, and the number of particles was counted using the "Analyze Particles" function of Fiji[78] with 0.1 to 4 μm² (CaLB1 and ALIX) and 10 μm² (autophagosomes). Range size and a circularity were set between 0-1. The number of puncta was normalized for 10,000 μm².

For condensate analyses in vivo, 5-day-old seedlings grown on 1/2 MS plates were transferred on a microscopy slide with either water, 5% 1,6-hexanediol (Merck, 240117), or 5% 1,2,6-hexanetriol (Merck, T66206) immediately before imaging. For the recovery after 1,6-hexanediol treatment, seedlings were left for 1 min in 5% 1,6-hexanediol and then placed for 15 min in liquid 1/2 MS supplemented with 150 mM NaCl. For counting the number of puncta, a threshold was set for each image, the images were converted to binary images, masked, and the number of particles was counted using the "Analyze Particles" function in Fiji with 0.05/0.1 to 4 μm² as size range and 0 to 1 circularity. The number of puncta was then normalized for the area of 10000 μm².

For E-64d treatment upon salt stress, seedlings were grown on 1/2 MS for 5 days and subsequently transferred to 1/2 MS plates supplemented with 100 mM NaCl for an additional 2 days. Seedlings were then treated with 100 μM E-64d (CaymanChem, 13533) for 2 to 4 h in liquid 1/2 MS and then for 10 min in phosphate buffered saline containing 50 μM MDC. For lanthanum(III)-chloride (LaCl₃, Merck, 262072) treatment, seedlings were incubated in 1/2 MS supplemented with 150 μM LaCl₃ for 1 or 2 h prior to imaging.

## Protoplast transformation

To analyze the number of GFP-ALIX puncta in protoplasts, protoplastation of root-derived cells was conducted as described previously[80]. In short, cells from a root cell culture were digested with an enzyme solution [0.4 M Mannitol, 5 mM EGTA supplement with 1% cellulase (100 mg) and 0.25% macerozyme (25 mg)] and incubated at room temperature in the dark for three hours while gently shaking. Subsequently, protoplasts were collected and washed with solution A (0.4 M Mannitol, 70 mM CaCl₂, 5 mM MES) and incubated in MMG solution (15 mM MgCl₂, 5 mM MES, 0.4 M Mannitol) for 30 min on ice. 100 μl of the protoplasts were used for transformation with 20 μg plasmid DNA. 400 μl PEG solution [40 % Polyethylenglycol (PEG) 4000, 0.4 M Mannitol, 0.1 M Ca(NO₃)] was added and after 30 min incubation on ice, protoplasts were washed with dilution solution (0.4 M Mannitol, 125 mM CaCl₂, 5 mM Glucose, 5 mM KCl, 1.5 mM MES) and collected by centrifugation. For the expression of the transformed constructs, protoplasts were incubated at 21 °C in the dark for 12–20 h in 4.6 g/L MS including Gamborg B5 vitamins medium and 0.4 M Mannitol. Cells were imaged using confocal laser microscopy.

The transformation of protoplasts obtained from wild-type and *calb1-1* seedling roots was performed as described before[81] with slight modifications. Instead of leaf material from 4-week-old plants, roots from 10 g of 10-day-old seedlings were cut into 0.5 mm pieces and digested with 20 mM MES (pH 5.7) containing 1.5% (w/v) cellulase 0.4% (w/v) macerozyme, 0.4 M mannitol, and 20 mM KCl. Root material was vacuum infiltrated for 30 min in the dark using a desiccator and incubated in the dark for 3 h at room temperature. Protoplasts were subsequently washed in W5 solution [2 mM MES (pH 5.7) containing 154 mM NaCl, 125 mM CaCl₂ and 5 mM KCl] and resuspend in MMG solution (4 mM MES (pH 5.7), 0.4 M mannitol, 15 mM MgCl2) for 30 min at room temperature. 100 μl of protoplasts were mixed with 20 μg DNA and 110 μl of PEG solution [40% (w/v) PEG4000, 0.2 M mannitol and 100 mM CaCl₂] and incubated for 15 min on room temperature. Cells were washed with W5 solution and incubated at 21 °C in the dark for 12–20 h in WI solution [4 mM MES (pH 5.7) containing 0.5 M mannitol, 20 mM KCl] until imaging.

## Condensates formation and FRAP analyses

Recombinant proteins were labeled with an excess of dye for 2 h at 4 °C and the free dye was removed by buffer exchange using Microcon® 10 K (Merck, MRCPRT010) or 30 K (Merck, MRCF0R030) centrifugal filters. ALIX was labeled with Alexa-488™ and CaLB1 and CaLB1(C2) were labeled with either NT-547 (Monolith NT Protein Labeling Kit GREEN-NHS MO-L002) or NT-RED 2ⁿᵈ generation dyes (NanoTemper, MO-L011).

Proteins were mixed in sodium phosphate buffer (39 mM Na₂HPO₄, 11 mM NaH₂PO₄, 150 mM NaCl, 10% glycerol, pH 7.5) with 5% polyethylene glycol (PEG) 8000. Before imaging, samples were placed into a 384 optical-well plate and incubated for 30 min at room temperature. For FRAP experiments, equimolar amounts of CaLB1 and ALIX were mixed, and condensates were bleached using the "bleach" function in ZEN black. After 10 images (pre-bleach), selected regions were bleached with 100 iterations at a laser power of 100%. Postbleach images were recorded every two seconds for a total time of 180 s. The fluorescence intensity/frame was measured using the plug-in Track-mate with a DoG detector and a Simple LaP tracker in Fiji[82]. The obtained values were analyzed with the online web tool easyFRAP (https://easyfrap.vmnet.upatras.gr)[83]. Fluorescent signal values from bleached condensates were used as the 'region of interest', values from unbleached condensates were used as the 'whole cell area', and values from the background used as the 'background'. The data were normalized using the full-scale method and fitted using the double exponential equation.

## RNA extraction and quantitative real-time (qRT)-PCRs

To examine the expression level of *CaLB1*, *ALIX*, *ATG8s*, and *CaLB1-GFP*, total RNA was isolated from 7-day-old seedlings with or without NaCl treatment using the NucleoSpin RNA Plant kit followed by cDNA synthesis using SuperScript II Reverse Transcriptase (Thermo Fisher, 18064014). The cDNA was analyzed using a CFX real-time PCR system (Bio-rad, 1855201) in combination with the iTaq Universal SYBR Green Supermix (Bio-rad, 1725120) with gene-specific primers TB575/TB472 for *CaLB1*, TB506/TB507 for *ALIX*, EI491/EI492 for *ATG8a*, EI493/EI494 for *ATG8b*, EI495/EI496 for *ATG8c*, EI497/EI498 for *ATG8d*, EI458/EI459 for *ATG8e*, EI460/EI461 for *ATG8f*, EI499/EI500 for *ATG8g*, EI501/EI502 for *ATG8h*, EI509/ EI510 for *ATG8i*, TB468/TB469 for *CaLB1-GFP*, and GP22/GP23 for *ACTIN8*.

## Generation of solid-supported lipid bilayers

POPC (1-palmitoyl-2-oleoyl-*sn*-glycero-3-phosphocholine, Merck, 850457 C), PI(3)P (1,2-dioleoyl-sn-glycero-3-phospho-(1′-myo-inositol-3′-phosphate, Merck, 850150P) and PI(4)P (1,2-dioleoyl-sn-glycero-3-phospho-(1′-myo-inositol-4′-phosphate, Merck, 850151 P). For POPC/PI(3)P or POPC/ PI(4)P membrane preparation, POPC was dissolved in chloroform (VWR, 22711.290) at 25 mg/ml, and PI(3)P and PI(4)P were dissolved at 0.5 mg/ml. 198 μl of POPC solution and 100 μl of PI(3)P or PI(4)P solution were mixed in a glass vial before being dried under a gentle nitrogen stream for 5 min and put in a vacuum chamber for two hours. After the removal of chloroform, the lipids were resuspended in 1 ml Tris-HCl (50 mM Tris, 100 mM NaCl, 10%(v) glycerol, pH 7.3) and were incubated at room temperature for 30 min. The resuspended solution was sonicated using a tip sonicator (Hielscher Ultrasound Technology) followed by extrusion using a handheld extruder from Avanti Polar Lipids (Merck, with a 30 nm polycarbonate membrane to produce small unilamellar vesicles (SUVs).

For the preparation of a solid-supported lipid bilayer (SSLB), the internal reflection element (IRE) of the ATR-cell, a silicon crystal, was polished using a smooth cloth and a 0.1 μm diamond polishing paste. Afterwards, the IRE was treated with H₂SO₄ (95%) for 10 min, followed

by rinsing with Milli-Q water and dried under a gentle nitrogen stream. This procedure was repeated 3 times to generate a hydrophilic surface and to induce the rupture of the SUVs and lipid bilayer formation on the IRE surface. 30 μl of the SUV solution were added to the IRE, and the formation of the bilayer was monitored spectroscopically for 40 min at 25 °C. Following this, the IRE was rinsed for 20 min using a buffer solution containing the same $Ca^{2+}$-concentration as for the measurements on protein interaction later, to remove leftover vesicles from the supernatant. 65 μl of protein solution containing 0.1 mg/ml CaLB1 dissolved in the same buffer that was used for the rinsing step, was added to the SSLB and spectra were recorded for 90 min. Afterwards, the ATR-crystal was rinsed for 40 min using Tris-HCl buffer containing the same $Ca^{2+}$-concentration that was used for the previous steps to confirm that the protein was bound to the membrane. Each measurement was repeated to validate the reproducibility with three replicated measurements being averaged.

### ATR-FTIR spectroscopy

FTIR spectra were recorded using a Vertex 70 v spectrometer (Bruker Optics, Germany) and a Bio-ATR II cell (Bruker Optics). The IRE consists of silicon with a refractive index of $n_1 = 3.42$ which results in a penetration depth of ~ 0.85 μm at 1000 $cm^{-1}$ for a sample with a refractive index of $n_2$ ~ 1.5 and a 45° angle of incidence of the IR beam. During the experiment, the flow rate of the peristaltic pump (Ismatec) was set 390 μl/min, and the ATR cell was set to 25 °C using a water bath with an integrated E300 immersion thermostat (LAUDA-Brinkmann). For each spectrum, 128 scans were recorded with a resolution of 4 $cm^{-1}$. For the analysis of the protein variants, background spectra were recorded after the formation of the SSLB. After the measurements were recorded, an Opus macro (Bruker Optics, Germany) was used to translate the spectral data to csv files. The amide II band area was integrated between 1600 $cm^{-1}$ and 1500 $cm^{-1}$ for each measurement with the derived values being averaged for the replicated measurements and plotted over time.

### Spin labeling and EPR spectroscopy

For spin labeling, 100 μM protein in PBS was incubated with 1 mM (1-oxyl-2,2,5,5-tetramethylpyrroline-3-methyl) methanethiosulfonate (MTSL) in a total reaction volume of 500 μL (2 h, 400 rpm, 20 °C). The excess spin label was removed by diafiltration with Amicon Ultra-0.5 mL Centrifugal Filters (3 K NMWL) and 10 × 400 μL PBS.

For continuous wave (CW) EPR spectroscopy, HIRSCHMANN ringcaps® capillaries (1.02 mm inner diameter) were filled with 25 μL spin-labeled protein sample and sealed with Hemato Seal™ capillary tube sealant (Fisher Scientific). Spectra were recorded for 10 min at the X band (9.645 GHz) on an EMXnano spectrometer (Bruker). The center field was set to 3435 G, the sweep width to 200 G, the modulation amplitude to 1 G, and the microwave attenuation to 15 dB.

For double electron-electron resonance (DEER) experiments, spin-labeled protein samples were incubated with or without 10 mM $CaCl_2$ (15 min, 4 °C) and subsequently mixed with 20 % (v/v) $d_8$-glycerol. 60 μL were filled into EPR tubes (fused quartz tubing, 2 mm inner diameter, Technical Glass Products) and flash-frozen in liquid nitrogen. DEER measurements were performed at Q-band (34 GHz) on an Elexsys E580 spectrometer (Bruker) equipped with an arbitrary waveform generator (AWG) unit (Bruker), a 150 W traveling-wave tube (TWT) amplifier (Applied System Engineering), and an overcoupled Q-band resonator (Bruker, ER5106QT-2). A cryogen-free helium recirculation system with a ColdEdge cryocooler (Bruker, CE-FLEX-4K-0110), a F-70H helium compressor (SHI cryogenics), and a MercuryiTC temperature controller (Oxford Instruments) were used to control and set the temperature to 50 K. All samples were measured using the standard four-pulse DEER sequence ($\pi_{obs}/2$ - $\tau_1$ - $\pi_{obs}$ - ($\tau_1 + t$) - $\pi_{pump}$ - ($\tau_2 + t$) - $\pi_{obs}$ – $\tau_2$–echo)[84] with rectangular observer pulses at 33.80 GHz and a rectangular pump pulse at 34.00 GHz. The interpulse

delay $\tau_1$ was set to 400 ns, the dipolar evolution time $\tau_2$ to 2000 ns, and the shot repetition time to 4080 μs. For nuclear modulation averaging, $\tau_1$ was increased in 8-steps by 16 ns each. Artefacts caused by interfering echoes were removed with an 8-step {(x) x [xp] x} phase cycling[85]. Procession and analysis of raw DEER data was performed with MatLab R2022a, the toolbox EasySpin 5.2.25[86] and ComparativeDeerAnalyzer within DeerAnalysis 2022[87].

### Transmission electron microscopy and immunolabeling

Root tips from 5-day-old seedlings were high-pressure frozen in a Leica EM ICE high-pressure freezer (Leica). The sections were imaged with the transmission electron microscope JEOL 2100 Plus (Jeol) operated either at 120 or 200 kV.

Samples used for immunogold-labeling were freeze-substituted in a Leica EM AFS2 (Leica) in an acetone solution containing 0.2% uranyl acetate, 0.2% glutaraldehyde, 0.6% ddH20 and 0.8% methanol[88]. Freeze substitution was conducted first at −90 °C for 116 h and, subsequently, the temperature was increased over 6 h to −60 °C. Root tips were then infiltrated with a methacrylate-based resin Lowicryl HM20 (Polysciences, Inc., Warrington, PA) and embedded in 100% HM20 in dry acetone. After placing 50 nm sections on Nickel grids, samples were immunolabeled. Grids were blocked at room temperature for 20 min in PBS supplemented with 0.01% Tween20 and 1% BSA and incubated for one hour with an anti-GFP antibody (1000×diluted, Abcam, ab13970). The grids were then washed and incubated with an anti-chicken antibody conjugated with 12 nm gold particles (10× diluted, Dianova, 703-205-155) for one hour at room temperature. After fixation in PBS containing 1% glutardialdehyde, samples were post-stained with 1% uranyl acetate (Merck) and subsequently with 0.2% Pb-citrate.

Samples used for the analysis of autophagosomes were freeze-substituted in a Leica EM AFS2 (Leica) in an acetone solution containing 2.5% OsO4. Freeze substitution was conducted first at −90 °C for 38 h. Subsequently, two cycles were started where the temperature was increased over 6 h of 30 °C and then kept stable for another 6 h. The two cycles were repeated twice to reach −30 °C. Temperature was then increased to 0 °C over 3 h and kept stable at 0 °C for an additional hour. Root tips were then washed in dry acetone and infiltrated with EPON epoxy resin (Fluka) in dry acetone. Samples were post-stained with UranyLess (Science Services) and 0.4% lead citrate in 0.2% NaOH.

### Statistics & reproducibility

No statistical method was used to predetermine sample size. Following data points were excluded from the analyses. Root length analyses: ungerminated or seedlings with arrested growth were excluded from all replicates. FRAP analyses: frames taken after 200 s were not analyzed to avoid effects of photobleaching occurring during imaging. ATR-IR Spectroscopy: data over 90 min were not included in the analysis. No randomization was applicable since there was no organization in experimental groups. The Investigators were not blinded to allocation during experiments and outcome assessment. Experiments were replicated three times to assure reproducibility, with exceptions indicated for individual experiments. The number of replicates and sample size are indicated in the figure caption for each experiment. Two-tailed $t$-test with no equal variance and two-tailed two proportion $z$-test were performed in Excel (Microsoft) as indicated for each experiment.

### Reporting summary

Further information on research design is available in the Nature Portfolio Reporting Summary linked to this article.

## Data availability

Data necessary to interpret, verify, and extend the research presented in this work are provided within the paper, the Supplementary data,

the Supplementary Information, and the Source data file. Sequence information for Arabidopsis was retrieved from TAIR (https://www.arabidopsis.org) and for other plant species from Phytozome (https://phytozome-next.jgi.doe.gov). The budding yeast Bro1 sequence was retrieved from SGD (https://www.yeastgenome.org), and the human ALIX sequence from GenBank (https://www.ncbi.nlm.nih.gov/genbank/). Structural models were obtained from AlphaFold II (https://alphafold.ebi.ac.uk). Source data are provided with this paper.

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

## Acknowledgements

We thank Kamila Kalinowska (Technical University of Munich at the time of experiments), Jens Werner, Yanna Piccini, and Lenard Kreis (University of Konstanz at the time of experiments) for earlier contribution to the project, Pascal Braun (Helmholtz Zentrum Munich) and Angela Alkofer (Technical University of Munich) for providing the Arabidopsis cDNA ORF library, Michael Laumann and Paavo Bergmann (Electron Microscopy Centre) for their expert help in electron microscopy, the gardeners of the botanical garden at the University of Konstanz for taking care of the plants, and the BioImaging Centre for the help in confocal microscopy. We thank all colleagues who shared published materials with us. We acknowledge support from the German Research Foundation (DFG) [SFB969 (E.I., F.S., and K.H.), INST 38/579-1 (E.I.) as well as projects 189682160 (E.I.) and 496470458 (F.S.)] and from the European Research Council (ERC) under the European Union's Horizon 2020 Research and Innovation program [grant no.772027; SPICE, ERC-2017-COG (M.D.)].

## Author contributions

N.M. and E.I. have designed the study and most of the experiments. N.M., N.S.L., T.B., and K.V. performed and analyzed protein biochemical assays, T.B., N.M., M.-K.N., and F.G. performed molecular cloning and phenotypic analyses, N.M., F.G., and M.-K.N. performed confocal microscopy, S.M. and K.H. designed and performed ATR-IR spectroscopy, M.L.N. and F.S. conducted and analyzed the FRAP experiments together with N.M., and E.L. and M.D. designed and carried out EPR spectroscopy analyses. K.M., N.M., and M.-K.N. prepared and analyzed electron micrographs. N.M. and E.I. wrote the manuscript with the input of all the other authors.

## Funding

## Competing interests

The authors declare no competing interests.
