## [Peer Review file · Nature Communications]

REVIEWER COMMENTS

Reviewer #1 (Remarks to the Author):

The authors of this manuscript report that CaLB interacts with ALIX and localizes with ATG8 on autophagosomes and that CaLB and ALIX undergo phase separation which could promote the recruitment of ESCRT-III subunit to autophagosomes. While the proposed model is attractive, the presented data is insufficient to support their conclusion. In particular, I found the conclusion “function of CaLB in enhancing phase separation of ALIX and facilitating the recruitment of ESCRT-III to the site of phagophore closure thereby ensuring efficient maturation of autophagosomes (Line 42-43)” is not well supported, most the results we have here are protein interaction, localization and mutant phenotype. No direct evidence is given related to phagophore closure, the requirement of phase separation in ESCRT recruitment, please find my point-to-point comments below:

(1) The result showed that the mRNA expression level of the *calb* CRISPR line is reduced significantly (Fig. 4a), which CRISPR line was detected here? And how the primers were designed for qPCR since both the two CRISPR lines generate truncations of CaLB? Please provide further information here, I may have missed this.

(2) Which *calb* CRISPR line was used to analyze the phenotype when seedlings were subjected to NaCl treatment in Fig. 4c? There were two CRISPR mutants generated in this study, so it would be good to include the data using two independent lines.

(3) The authors found that the numbers of CaLB-GFP and GFP-ALIX labelled puncta increased following NaCl treatment and concluded that CaLB and ALIX could be associated with salt-induced autophagosomes (lines 181-183, Fig. 5a-d), the CaLB-GFP or GFP-ALIX labelled puncta do not represent autophagosomes unless they colocalize with ATG8. Therefore, this statement is not accurate, it can only say that CaLB or ALIX puncta is induced.

(4) The mRNA levels of ALIX and ATG8 did not change upon NaCl treatment (Fig. 5e and Fig. S6b), Will there be any difference in protein levels of GFP-ALIX or ATG8 after salt treatment, as the image showed that the fluorescence intensity of GFP-ATG8a seems to be much lower with NaCl treatment (Fig. 5b)?

(5) The authors showed the colocalization of CaLB-GFP and RFP-ATG8a after NaCl treatment in Fig. 5g. Will more CaLB-GFP be localized to RFP-ATG8a labelled autophagosomes upon NaCl treatment compared to that without NaCl treatment? It is essential to prove that the recruitment of CaLB to autophagosomes is in response to salt stress. Similar analysis should also be conducted in plants expressing ALIX-GFP and RFP-ATG8i.

(6) The authors indicated that “CaLB-GFP signals remained mostly in the cytosol and the signal intensity was weaker in the vacuole compared with the cytosol, indicating that CaLB is not as efficiently targeted to the vacuole as the ATG8s” in lines 228-230 (Fig. S7d-e), however, without any quantification analysis, the evidence is not sufficient to draw this conclusion. Besides, a better way to identify whether the vacuolar turnover of CaLB-GFP is weaker than ATG8s is to detect the protein

levels of CaLB-GFP following certain treatments and calculate the free-GFP/full-length ratio compared to that of ATG8s. The same problem also appears in Fig. S7f.

(7) In Fig. 8a-b, the authors try to prove that the calb CRISPR mutant blocks the transport of autophagosomes into the vacuole, however, the evidence is too weak. Do Col-0 and the calb mutant generate similar amounts of autophagosomes in cytosol upon NaCl treatment? Is there any difference in the ratios of vacuole/cytosol autophagosomes between Col-0 and the calb mutant under salt stress? In addition, the root regions used in Fig. 8a-b seem not to be consistent with those in other figures (e.g. Fig. S7).

(8) The representative images are missing for the quantification analysis in Fig. 8g-i.

(9) The subtitle in line 272 “CaLB and ALIX modulate autophagy by recruiting ESCRT to the autophagosomes” is not convincing, as the authors did not show any results on the function of ALIX in modulating autophagy.

(10) It is proposed that CaLB triggers the assembly of ALIX into molecular condensates and localization to autophagosomes without direct evidence. Will the phase separation of ALIX be blocked in calb mutant? The co-localization of ALIX and ATG8 should be analyzed in the calb mutant to show the effect of CaLB on ALIX localizing to autophagosomes.

(11) The authors indicate that “phase separation of ALIX and CaLB mediated through their PRDs”, there should be some endogenous evidences to support this point. For example, the authors can examine the ALIX phase separation as well as its localization to autophagosomes in the calb/CaLB (ΔPRD) background.

(12) Overall, some of the statements related to Figure 8 are over-interpreted, and the model in Figure 8j is not well supported. As indicated in Figure 5, only 15% of CaLB is associated with ATG8-labelled autophagosomes, it is not possible to say CaLB is involved in autophagosome maturation and membrane scission (Lin 297-304).

Reviewer #2 (Remarks to the Author):

In the manuscript "Phase separation of Arabidopsis CaLB modulates autophagosome maturation by ALIX3 dependent ESCRT recruitment" Mosesso et al. describe CaLB protein found to interact with the ALIX during salt stress-induced autophagy. Authors propose that CaLB recruits ALIX and subsequently ESCRTIII to the maturing autophagosomes thereby facilitating closure of autophagosomal membrane.

The study comprises an impressive set of experiments showcasing interesting and significant findings. However, for the sake of clarity and more constructive presentation of the results, the manuscript itself might require some major and minor changes.

The comments in the sequence of their occurrence:

1. Firstly, the title does not fully represent the findings of the study, in which authors clearly demonstrate that CaLB partakes in a specific type of autophagy (salt stress-induced autophagy) and does not impact autophagy triggered by carbon deprivation.

Additionally the introduction, the final scheme and the discussion do not focus on the salt stress response and therefore together with the title provide a confusing message: it becomes unclear whether authors propose the mechanistic insight for salt-stress induced autophagy or for autophagy in general.

2. The introduction is very nicely written but might benefit from more information on plant CaLB proteins to explain implemented mutagenesis and also lay the ground for salt-stress related autophagy.

3. In the Fig.1A and B not all abbreviations are explained. Why are aa 540 and 630 indicated on the ALIX scheme? The main text could benefit from a more detailed description of the protein domains and their functions and the logic behind the performed mutagenesis. That would also make it easier to understand the application of the yeast ortholog carrying the name of the Arabidopsis domain (Lines 129-130).

4. Could authors elaborate on at least suggested nature of the foci used for quantifying colocalization of ALIX and CaLB (Fig.1G)? It is not clear why foci were selected for co-localization. Maybe authors would like to consider Manders method for co-localization (doi: 10.1152/ajpcell.00462.2010, see ImageJ plugin JACoP) that would provide information on proportion of ALIX and CaLB co-localizing with each other as well as statistical significance of detected co-localizations.

5. Fig 1FB: Y2H proteins are plant orthologs only?

6. What is the reason for indicating CBR1-3 on the Fig2A,B, maybe it would be good to have some statement in the main text reflecting on the importance of these regions for the assays? Could

authors state why not the complete protein was used for the assay in Fig 2 FB? Does IDR lead to precipitation?

It is not clear how the values were normalized for Fig 2C and why the y axis starts from 732.8%, permille? It is therefore difficult to understand why the values for D32A are so much lower than for WT with Mg²⁺.

7. We strongly recommend that authors reconsider the layout of the panels for lipid binding assays. The current Fig2D looks startling, like made up results. Although, intention of authors to save space on the figure is understandable, the labelling for lipid droplets positions would be most informative directly by the corresponding panels.

Fig 2F legend states that values were plotted from 2 to 3 experiments-does this mean that different number of experiments were used for different conditions and if yes, why are the all plotted together on the same chart?

8. (Minor comment) Please note that the panel labels on the figure 3 do not match the figure legend/text.

9. Fig 3D From the current description it is not particularly clear how authors selected the amino acids for mutagenesis. Maybe Fig 3 could be supplemented with an AlphaFold predicted structure of CaLB illustrating position of amino acids subjected to mutation, to visualize that those are present on the protein surface and available for interaction with ATG8.

10. Lines 161 and onwards: as mentioned above, providing more elaborate background on CaLB protein in the intro would make the choice of salt stress more intuitive for the reader. Currently the choice of stress trigger is not clear. For the sake of clarity, authors would probably want to rephrase broad statements about the role of CaLB in autophagy. If CaLB is involved in all autophagosome biogenesis, why do calb knockout mutants not show susceptibility to carbon starvation, but only to salt stress?

11. Fig 4A and D, authors might want to double check the significance level of the observed differences. Additionally, the chart in Fig4D contains only one series, while it is annotated as comparison of root lengths with and without NaCl.

12. Figure 4 should probably also contain side by side comparison of CaLB and ATG knockout seedlings sensitivity to NaCl?

13. It would be most instructive to see if introducing calb mutation into ATG-deficient background provides no additional phenotypes. However, establishing such lines would cause an unreasonable delay for the publication and is not recommended if the required material is not available.

14. The scale bar size is missing for the Fig 5A-C

15. For the Figure 5 it would be instructive to indicate what was counted as puncta and why (maybe an illustrative example could be shown in a magnified inset with arrowheads pointing at puncta).

16. Fig 5d might have a mistake in y axis labelling. Judging by Figs5a-c, it is unlikely the authors observed 40 autophagosomes/10 square um or is it 10 000 um²?

17. Fig 5e is not sufficiently described in the text to understand authors opinion: why does expression of CaLB go down after 90 minutes? How is this related to CaLB role in autophagosome biogenesis? In the FigS6b the ATG8e expression pattern seems to closely resemble that of CaLB.

18. The way the co-localization of ALIX and CaLB to autophagosomes is presented does not align very well with the fractionation experiments shown on the same figure.

It would be informative to include co-localization of CaLB and ALIX under the same salt stress conditions.

The relevance of the localization phenotypes quantified in Fig 5j and n are not discussed, furthermore, ring-like localization of ALIX seems to contradict the suggested mechanism on the final scheme. Possibly, the charts with colocalization phenotypes are better suited for supplementary figures/future perspective?

19. Authors do not explain their choice of ATG8 orthologs for different experiments shown in Figure 5. For example, why is panel g showing ATG8a and panel k is showing ATG8i?

20. Lines 223-227: it is a bit surprising that authors decided to use conditions that they previously show to cause no phenotype in the calb loss-of-function mutant. Is it not clear why this experiment is mentioned together with the NaCl treatment. Do authors see significance in this expectedly negative result? If yes, could authors elaborate on it?

21. To aid interpretation of the data, panels c and d in the Figure 6 require presentation of individual channels in addition to the merged image. It would be also great to have zoomed-in scans of autophagosomes, similar to those shown in Fig 5j. Furthermore, panel illustrates side by side micrographs made on cortical cytoplasm, containing a lot of puncta and micrograph made on the optical section including large empty vacuole. The latter would naturally contain less puncta and thus quantification per total area might be skewed.

This figure should also include similar assessment performed using ALIX marker line.

22. Lines 276-277: MDC has been demonstrated to be a questionable method for autophagy monitoring (<https://doi.org/10.1093/pcp/pcu041>, doi: 10.1080/15548627.2020.1797280.) It has to be performed with appropriate controls, including staining of ATG knockouts as a negative control and showing co-localization of GFP-ATG8 and MDC in the WT background under used conditions.

Additionally, measuring area of a 3D cluster of potentially autophagic bodies be not representative of autophagic activity. Authors might want to double check statistical significance of values show in Fig. 8b.

Reviewer #3 (Remarks to the Author):

Mosesso et al. identified a new factor that interacts with ALIX via yeast two-hybrid screen and confirmed their interaction. They demonstrated that as the name suggests, CaLB binds to lipids in a calcium-dependent manner. They also found that CaLB interacts with ATG8 and localizes on salt-induced autophagosomes. The authors further showed that CaLB co-phase separates with ALIX in vitro and forms condensates in vivo. Finally, they showed that CaLB is required for the recruitment of ESCRT-III to autophagosomes.

The findings of this work provide insights into how ESCRT is recruited to autophagosome to help maturation, in particular, under stress conditions. The CaLB-ALIX module could be one of the routes that relay stress signal to protein quality control via autophagy. However, based on current data, I am concerned about the significance of CaLB-ALIX module for the following reasons. As the author also mentioned in the discussion, others pathways such as FREE1-mediated recruitment of ESCRT-III to autophagosome also exist.

The importance of CaLB in salt stress tolerance is questionable. The authors stated that mutant seedlings showed a significant reduction of primary root length compared with the wild type but the photo shown in Figure 4c seems not supporting this conclusion. I did a rough measurement, the resulting normalized ratio is similar in wild type and mutant. In any case, the current does not support the necessity of CaLB for salt stress tolerance.

The role of CaLB in the efficient maturation of autophagosomes is rather speculative. Electron microscopic experiments should be performed to directly observe whether autophagosome are immature in *calb* mutant background.

As the title suggests that phase separation of CaLB modulates autophagosome maturation, but there is no evidence support this. CaLB may phase separate *in vivo*, but genetic evidence disrupting phase separation is required to prove whether phase separation is necessary for its function in autophagosome maturation.

The authors used suspension cells in which CaLB is transiently knocked down, the efficiency of which is low, to test the colocalization of ESCRT with autophagosomes. As we know that the concentration is key for phase separation, but all proteins were overexpressed. The conclusions drawn from these data are not convincing. These experiments should be done in CRISPR mutant background.

In line with my concern, I find it difficult to understand that knock down of CaLB increased the number of autophagosomes per cell and more autophagosomes colocalized with ESCRT-III, but the authors think phase separation of ALIX and CaLB on the autophagosomal membrane could facilitate ESCRT-III recruitment. It is also overstated to say CaLB phase separation could facilitate membrane scission.

The increased number of CaLB-GFP and GFP-ALIX foci correlated with induced CaLB transcript level by salt, but it is not clear whether CaLB expression is the cause. As it is a calcium-dependent lipid binding, to test whether the increase of intracellular foci positive for CaLB-GFP, GFP-ATG8a, and GFP-ALIX under salt stress is due to calcium, the authors can use calcium channel inhibitors such as LaCl₃.

How does the authors define CaLB-GFP signals that are proximal to mRFP-ATG8 signal shown in Figure 5g, they might be just close to each other, not necessarily associating, as from the biophysical perspective, the association of a condensate with similar size to the autophagosome to the autophagosomal membrane is unfavorable. The data showed that 15.0%, 8.3%, and 15.5% of structures positive for ATG8a, ATG8e and ATG8i, respectively, colocalized with CaLB-GFP, which is

not good colocalization, leading to the conclusion that CaLB localizes to autophagosome, but when 16.6% of CaLB-GFP foci co-localized with ARA6-mRFP in the presence of Wortmannin, the authors concluded that CaLB does not localize on endosomes with high frequency.

Higher resolution is required to show 35.3% of CaLB-mRFP foci colocalize with GFP-ALIX.

Why the authors used BFA and WM when observing colocalization of CaLB with SYP43 and ARA, respectively? Intuitively, the observation of native endosomes does not need these drugs, therefore, the conclusion that that CaLB does not localize on endosomes with high frequency is not confident.

The quality of immuno-electron microscopy on CaLB-GFP is too poor. In theory, it is possible to see a high density region around the gold particle signal if CaLB does form condensates on autophagosome membrane.

When observing ALIX localization on autophagosomes, the authors were able to see the ring-like structures, but when observing CaLB, ATG8 signal is always a filled structure, is it a matter of resolution or other reasons. Also, better quality images are needed to convince me that ALIX also form puncta similarly to CaLB on autophagosome.

When interrogating the property of CaLB condensates, 5% 1,6-hexanediol is already very strong, it is not enough to judge if they are liquid-like or not only based on the dissolution. A further experiment could be wash out of the hexanediol to see whether CaLB condensates reform in the same spot. FRAP experiment should also be performed.

The in vitro data support that CaLB enhances the phase separation of ALIX, the authors should also test whether the condensation of ALIX in vivo depends on CaLB. This should be straightforward to observe in the CRISPR mutant background.

For the FRAP experiment in vitro, selected regions were bleached with 100 iterations at a laser power of 100%, I think this could be over-bleached. Sometimes, overbleach leads to less dynamic and recovery.

REVIEWER COMMENTS

We sincerely thank all three reviewers for taking their time to read our manuscript and for their valuable suggestions and constructive comments to improve the manuscript. We have carefully read and considered all comments and adjusted the manuscript title, added controls, and performed a number of additional experiments that are now included in the revised manuscript. Where necessary, we have corrected the manuscript according to the reviewers' suggestions. The point-by-point answers to the reviewers' comments can be found below.

As "CaLB" refers to a broad protein family containing C2 domains, to avoid confusions, we have named the protein coded by At4g34150 as "CaLB1".

Reviewer #1 (Remarks to the Author):

The authors of this manuscript report that CaLB interacts with ALIX and localizes with ATG8 on autophagosomes and that CaLB and ALIX undergo phase separation which could promote the recruitment of ESCRT-III subunit to autophagosomes. While the proposed model is attractive, the presented data is insufficient to support their conclusion. In particular, I found the conclusion "function of CaLB in enhancing phase separation of ALIX and facilitating the recruitment of ESCRT-III to the site of phagophore closure thereby ensuring efficient maturation of autophagosomes (Line 42-43)" is not well supported, most the results we have here are protein interaction, localization and mutant phenotype. No direct evidence is given related to phagophore closure, the requirement of phase separation in ESCRT recruitment, please find my comments below:

>We thank the reviewer for the helpful comments and suggestions. We have now performed several additional experiments that show that CaLB1 is required for the proper maturation and delivery of autophagosomes. Please find our point-by-point response below.

(1) The result showed that the mRNA expression level of the *calb* CRISPR line is reduced significantly (Fig. 4a), which CRISPR line was detected here? And how the primers were designed for qPCR since both the two CRISPR lines generate truncations of CaLB? Please provide further information here, I may have missed this.

> The qRT-PCR was performed with the *calb1-1* line shown in Supplemental Figure 4d. We have now also added the result for *calb1-2* in Supplemental Figure 4e. The primer listed in the previous submission had an overlap with the mutated sites. We therefore redesigned a primer binding after the indel mutations and performed the qRT-PCR analysis again. The position of the primers is indicated now in the new Supplementary Figure 4b.

Both CRISPR events lead to the appearance of premature stop codons in the mRNA, which could lead to the production of truncated proteins if these are translated. However, the CRISPR event should not interfere with the transcription, and thus in principle the mRNA should be transcribed as in the wild type with the respective mutations.

Unfunctional mRNAs with unintended stop codons are subject to the non-sense-mediated mRNA decay pathway which is conserved in eukaryotic organisms [Causier et al (2017)

Scientific Reports). We assume that the *CALB1* mRNAs with premature stop codons caused by the CRISPR event was unstable due to this mechanism.

(2) Which *calb* CRISPR line was used to analyze the phenotype when seedlings were subjected to NaCl treatment in Fig. 4c? There were two CRISPR mutants generated in this study, so it would be good to include the data using two independent lines.

> The analysis shown in Figure 4c was performed with the *calb1-1* line. The second CRISPR line (*calb1-2*) is now included in Supplementary Figure 4i and j. Both lines show salt sensitivity (lower ratio of primary root length with NaCl/without NaCl), and complementation of this phenotype in *calb1-1* with the own promoter driven *CaLB1pro:CaLB1-GFP* shows that the phenotype is a consequence of the mutation in the *CaLB1* locus.

(3) The authors found that the numbers of CaLB-GFP and GFP-ALIX labelled puncta increased following NaCl treatment and concluded that CaLB and ALIX could be associated with salt-induced autophagosomes (lines 181-183, Fig. 5a-d), the CaLB-GFP or GFP-ALIX labelled puncta do not represent autophagosomes unless they colocalize with ATG8. Therefore, this statement is not accurate, it can only say that CaLB or ALIX puncta is induced.

> Thank you for pointing this out. We have now rephrased the sentence accordingly.

(4) The mRNA levels of ALIX and ATG8 did not change upon NaCl treatment (Fig. 5e and Fig. S6b), Will there be any difference in protein levels of GFP-ALIX or ATG8 after salt treatment, as the image showed that the fluorescence intensity of GFP-ATG8a seems to be much lower with NaCl treatment (Fig. 5b)?

>Following the reviewer's suggestion, we have now included an immunoblot analyzing the protein levels of CaLB1-GFP, GFP-ALIX, and GFP-ATG8a with and without the addition of salt. The results are shown in Figure 4b and indicates that all three proteins accumulate upon salt treatment. The fluorescence intensity of GFP-ATG8a in Figure 5b is much less pronounced in the cytosol after salt treatment, but clearly accumulates in cytosolic foci that increase in number as shown in Figure 5d. The increase in the number of foci positive for CaLB1-GFP, GFP-ALIX and GFP-ATG8a coincides with the stabilization of these proteins.

(5) The authors showed the colocalization of CaLB-GFP and RFP-ATG8a after NaCl treatment in Fig. 5g. Will more CaLB-GFP be localized to RFP-ATG8a labelled autophagosomes upon NaCl treatment compared to that without NaCl treatment? It is essential to prove that the recruitment of CaLB to autophagosomes is in response to salt stress. Similar analysis should also be conducted in plants expressing ALIX-GFP and RFP-ATG8i.

>We analyzed the colocalization efficiency of CaLB1 and ALIX with ATG8i and included this percentage in the main text (lines 246-254&285-290). The colocalization efficiency per se did not change dramatically, however, given the overall increase in the number of autophagosomes upon salt treatment, the absolute number of autophagosomes with ALIX and CaLB1 signals is increased.

(6) The authors indicated that “CaLB-GFP signals remained mostly in the cytosol and the signal intensity was weaker in the vacuole compared with the cytosol, indicating that CaLB is not as efficiently targeted to the vacuole as the ATG8s” in lines 228-230 (Fig. S7d-e), however, without any quantification analysis, the evidence is not sufficient to draw this conclusion. Besides, a better way to identify whether the vacuolar turnover of CaLB-GFP is weaker than ATG8s is to detect the protein levels of CaLB-GFP following certain treatments and calculate the free-GFP/full-length ratio compared to that of ATG8s. The same problem also appears in Fig. S7f.

>We agree with the reviewer that a quantitative analysis is required. We have therefore now included a further control experiment for the MDC staining (Supplemental Figure 8a and b) and performed a GFP cleavage assay (Figure 8b and c).

The GFP cleavage assay shows that free GFP is not clearly detectable in CaLB1-GFP or GFP-ALIX expressing lines even after E64-d treatment, in contrast to NBR1 or GFP-ATG8a shown in Figure 8c, j and k. This result suggests that both CaLB1-GFP and GFP-ALIX are not efficiently transported or degraded in the vacuole as expected from a structural component of the autophagosome or an autophagy cargo adaptor.

(7) In Fig. 8a-b, the authors try to prove that the *calb1* CRISPR mutant blocks the transport of autophagosomes into the vacuole, however, the evidence is too weak. Do Col-0 and the *calb* mutant generate similar amounts of autophagosomes in cytosol upon NaCl treatment? Is there any difference in the ratios of vacuole/cytosol autophagosomes between Col-0 and the *calb* mutant under salt stress? In addition, the root regions used in Fig. 8a-b seem not to be consistent with those in other figures (e.g. Fig. S7).

>We thank the reviewer for this comment. We carried out following additional experiments to strengthen the statement. The root regions used in Fig. 8a and b are both from the transition zone of the root. The original Supplementary Figure 7b showed more elongated cells compared to those in Figure 8. We went through the images and made sure that the images are taken from the approximate same region of the roots.

a. We performed qRT-PCR on all nine *ATG8s* upon salt treatment and showed that there is no significant difference between the wild type and the *calb* mutant (Supplemental Figure 8c).

b. We isolated autophagosomes from wild type and *calb1-1* mutant seedlings and performed a proteinase K protection assay (Figure 8i). The results show that GFP-ATG8a is less protected in the *calb1-1* mutant when compared with the wild type.

c. We monitored the cleavage of GFP from GFP-ATG8a in wild type and *calb1-1* mutant seedlings upon salt treatment (Figure 8j and k). There is less release of free GFP in the *calb1-1* mutant compared with the wild type, suggesting that there is less efficient transport/degradation of GFP-ATG8 containing autophagosomes in the *calb1-1* mutant seedlings.

d. We quantified the number of GFP-ATG8a-positive foci in wild type and *calb1-1*. Upon salt treatment, *calb1-1* has a higher induction in GFP-ATG8a foci (ratio with/without NaCl) (Figure 8e).

(8) The representative images are missing for the quantification analysis in Fig. 8g-i.

>We have now added experiments in stable Arabidopsis lines and have removed this data.

(9) The subtitle in line 272 “CaLB and ALIX modulate autophagy by recruiting ESCRT to the autophagosomes” is not convincing, as the authors did not show any results on the function of ALIX in modulating autophagy.

>As alix null mutants show seedling lethality, in depth analyses of autophagy in the alix mutant is difficult. Nonetheless, two independent alix mutant alleles show accumulation of ATG8 and NBR1, which was the starting point of this study. We now included the data in Figure 1a.

(10) It is proposed that CaLB triggers the assembly of ALIX into molecular condensates and localization to autophagosomes without direct evidence. Will the phase separation of ALIX be blocked in *calb* mutant? ALIX phase separate on its own-in vitro there is an enhancement when CaLB is present. The co-localization of ALIX and ATG8 should be analyzed in the *calb* mutant to show the effect of CaLB on ALIX localizing to autophagosomes.

>We agree with the reviewer that the analysis of ALIX and ATG8 colocalization *in planta* will provide an important insight. Despite our repeated efforts to combine the published GFP-ALIX line (Cardona-Lopez et al, 2015, The Plant Cell) with the *calb1* mutants generated in this study, we could not obtain homozygous *calb1* mutant lines expressing GFP-ALIX. We suspect this might be due to genetic linkage of the *calb1* locus and the GFP-ALIX transgene.

We next transformed the published *ALIXpro:GFP-ALIX* plasmid to the *calb1-1* homozygous mutants. However, even after screening all T1 lines we obtained, we could not identify a line that expresses ALIX to a level detectable at the confocal microscope. The available published GFP-ALIX line can be used for microscopic analyses optimally when the GFP-ALIX transgene is homozygous. We thus suspect that the own promoter-driven ALIX line has a very weak expression, and thus we were not able to identify an expressing line in the T1 generation.

To study the behavior of ALIX *in vivo*, we therefore used Arabidopsis root-derived cell culture and expressed GFP-ALIX together with a CRISPR construct that targets *CaLB1*. As a control, a mutated CRISPR construct was used. We have analyzed over 100 cells in each combination and the quantification of the images shows that in the presence of the CRISPR construct targeting *CaLB1*, the number of ALIX foci per cell is significantly reduced (Figure 6i and j).

(11) The authors indicate that “phase separation of ALIX and CaLB mediated through their PRDs”, there should be some endogenous evidences to support this point. For example, the authors can examine the ALIX phase separation as well as its localization to autophagosomes in the *calb*/*CaLB* (Δ PRD) background.

>We thank the reviewer for this suggestion. As mentioned above, most probably due to genetic linkage with the published ALIX-GFP line and the *calb1* mutants, we were not able to generate a *calb1* mutant line expressing ALIX-GFP. The limited time was not enough to generate new ALIX-GFP constructs and combine it with the mutant and a CaLB1(Δ PRD)

(12) Overall, some of the statements related to Figure 8 are over-interpreted, and the model in Figure 8j is not well supported. As indicated in Figure 5, only 15% of CaLB is associated with ATG8-labelled autophagosomes, it is not possible to say CaLB is involved in autophagosome maturation and membrane scission (Lin 297-304).

> We thank the reviewer for pointing this out. To address this point, we have now performed several additional experiments, which we believe supports and strengthens our model.

We performed centrifuge-based enrichment of autophagosomes and treatment with proteinase K, which is used as a method to assess the closure status of autophagosomes (Zhou et al, 2019, JCB).

About the relatively low colocalization efficiency: Our data suggests that CaLB1 is not constantly associated with autophagosomes, which is visible also from the time lapse experiment (Supplementary Figure 5c, d and e). In addition, as CaLB1 is not overall the autophagosomal membrane, depending on the confocal plane, we may miss the CaLB-GFP puncta on the autophagosome. We carefully went through the passage in the manuscript describing Figure 8 and modified the statement wherever necessary.

We believe that the new experiments and analyses performed according to the reviewers' suggestions strengthens the conclusion and model shown in Figure 9.

Reviewer #2 (Remarks to the Author):

In the manuscript "Phase separation of Arabidopsis CaLB modulates autophagosome maturation by ALIX₃ dependent ESCRT recruitment" Mosesso et al. describe CaLB protein found to interact with the ALIX during salt stress-induced autophagy. Authors propose that CaLB recruits ALIX and subsequently ESCRTIII to the maturing autophagosomes thereby facilitating closure of autophagosomal membrane.

The study comprises an impressive set of experiments showcasing interesting and significant findings. However, for the sake of clarity and more constructive presentation of the results, the manuscript itself might require some major and minor changes. The comments in the sequence of their occurrence

>We thank the reviewer for the insightful comments and constructive suggestions. The point-by-point answer to the points raised by the reviewer can be found below:

1. Firstly, the title does not fully represent the findings of the study, in which authors clearly demonstrate that CaLB partakes in a specific type of autophagy (salt stress-induced autophagy) and does not impact autophagy triggered by carbon deprivation. Additionally the introduction, the final scheme and the discussion do not focus on the salt stress response and therefore together with the title provide a confusing message: it becomes unclear whether authors propose the mechanistic insight for **salt-stress** induced autophagy or for autophagy in general.

>We thank the reviewer for the suggestion. In the revised manuscript, we have changed the title accordingly. We cannot rule out that CaLB1 functions also in other types of autophagy, however, focused the story on salt-induced autophagy rephrased the text.

2. The introduction is very nicely written but might benefit from more information on plant CaLB proteins to explain implemented mutagenesis and also lay the ground for salt-stress related autophagy.

>We thank the reviewer for the suggestion. In the revised manuscript, we have added a paragraph introducing plant CaLB proteins and salt-stress triggered autophagy now in the introduction section.

3. In the Fig.1A and B not all abbreviations are explained. Why are aa 540 and 630 indicated on the ALIX scheme? The main text could benefit from a more detailed description of the protein domains and their functions and the logic behind the performed mutagenesis. That would also make it easier to understand the application of the yeast ortholog carrying the name of the Arabidopsis domain (Lines 129-130).

>Thank you for pointing this out. We have now corrected the description in the scheme and added more information about the protein and protein domains.

4. Could authors elaborate on at least suggested nature of the foci used for quantifying colocalization of ALIX and CaLB (Fig.1G)? It is not clear why foci were selected for co-localization. Maybe authors would like to consider Manders method for co-localization (doi: 10.1152/ajpcell.00462.2010, see ImageJ plugin JACoP) that would provide information on proportion of ALIX and CaLB co-localizing with each other as well as statistical significance of detected co-localizations.

>We thank the reviewer for this suggestion. The colocalization of ALIX and CaLB₁ was counted manually without applying additional filters or thresholding. We also considered colocalization analysis using methods to calculate Mander's overlap coefficient. However, though this method works very well for other samples, for the particular experiments shown in this manuscript the methods do not seem to reflect the behavior of the fluorophore-fused proteins due to the high cytosolic signal and the small size of the foci. The cytosolic signals make the thresholding very difficult as applying a threshold value diminishing the cytosolic signals dramatically reduces the number of foci included in the analyses. Thus, although we are aware that the application of an automatized method is ideal, we opted to count the number of colocalizing foci manually.

5. Fig 1FB: Y2H proteins are plant orthologs only?

>The Y2H was performed only with plant proteins. To avoid misunderstandings, we now clarified this in the figure legend.

6. What is the reason for indicating CBR₁₋₃ on the Fig2A, B, maybe it would be good to have some statement in the main text reflecting on the importance of these regions for the assays? Could authors state why not the complete protein was used for the assay in Fig 2 FB? Does IDR lead to precipitation? It is not clear how the values were normalized for Fig 2C and why the y axis starts from 732.8%, permille? It is therefore difficult to understand why the values for D₃₂A are so much lower than for WT with Mg²⁺.

>We thank the reviewer for these comments. We have now clarified the description of sentence describing the CBRs (lines 125-127). Since Figure 2a and b were redundant, we only left Figure 2a to show the positions of the calcium-binding loops.

The C₂ domain of CaLB₁ was used for the MST analysis in Figure 3b (we assume Fig2 FB refers to this figure panel), as the full-length protein could not be purified at the concentration required to perform the analysis. We have now added this explanation in the text (lines 128-130).

The Y axis values indicate the normalized fluorescence [arbitrary unit], we have now corrected this in Figure 2. Since the fluorophore labelling efficiency is different from experiment to experiment and also varies between protein samples, the wild type CaLB₁ protein and the D₃₂A variant have different fluorescence intensity even the same amount of protein (50 nM) was used. Thus, the D₃₂A data points have a different starting point. The core message of the MST analysis is the ligand concentration-dependent change in fluorescence. As the wild type protein with Ca²⁺ clearly show a change, but the D₃₂A variant not, we conclude that the D₃₂A variant does not interact with Ca²⁺.

7. We strongly recommend that authors reconsider the layout of the panels for lipid binding assays. The current Fig2D looks startling, like made up results. Although, intention of authors to save space on the figure is understandable, the labelling for lipid droplets positions would be most informative directly by the corresponding panels. Fig 2F legend states that values were plotted from 2 to 3 experiments-does this mean that different number of experiments were used for different conditions and if yes, why are the all plotted together on the same chart?

>We apologize for the insufficient description of the experiment and confusion.

We removed the illustration of the lipid spotted membrane which was originally shown in Figure 2d. In the revised Figure 2c and Figure 2e, we show the scheme to the left of each experiment as suggested by the reviewer.

In the revised version of Figure 2d (formerly Figure 2f), the graph that shows the average and standard deviation of three experiments for all conditions [CaLB1(WT)+PI(3)P+20 μ M Ca²⁺, CaLB1(WT)+PI(4)P+20 μ M Ca²⁺, CaLB1(D32A)+ PI(3)P+20 μ M Ca²⁺].

In addition, to make the presentation clearer, we removed the membrane binding experiments of CaLB1 under different Ca²⁺ concentrations as we have moved the data into a meanwhile published method paper describing the membrane preparation and the ATR-IR spectroscopy analysis which [Maguire et al. (2023), Spectrochim Acta A Mol Biomol Spectrosc.].

8. (Minor comment) Please note that the panel labels on the figure 3 do not match the figure legend/text.

>Thank you for pointing this out. We have now corrected the labels.

9. Fig 3D From the current description it is not particularly clear how authors selected the amino acids for mutagenesis. Maybe Fig 3 could be supplemented with an AlphaFold predicted structure of CaLB illustrating position of amino acids subjected to mutation, to visualize that those are present on the protein surface and available for interaction with ATG8.

>Thank you for this comment. We have now included an Alphafold model of CaLB1 with the respective mutations highlighted (Supplementary Figure 3e).

10. Lines 161 and onwards: as mentioned above, providing more elaborate background on CaLB protein in the intro would make the choice of salt stress more intuitive for the reader. Currently the choice of stress trigger is not clear. For the sake of clarity, authors would probably want to rephrase broad statements about the role of CaLB in autophagy. If CaLB is involved in all autophagosome biogenesis, why do calb knockout mutants not show susceptibility to carbon starvation, but only to salt stress?

>Salt treatment was used as trigger as expression analysis of CaLB₁ as found in ePlant (<https://bar.utoronto.ca/eplant/>) showed induction of CaLB₁ upon salt treatment. We believe that CaLB₁ function becomes especially important upon salt stress, however, cannot rule out that CaLB₁ functions also in autophagy induced by other conditions. The GFP cleavage assay shows that the effect of *calb1* mutation is also present without salt treatment, suggesting CaLB₁ might be involved in broader autophagic processes. One explanation could be that other factors such as the recently published FYVE1/FREE1 (Zeng et al. 2023 Nat. Commun.) is functioning in parallel and have a prominent role, which could be why *calb1* mutants do not show susceptibility to carbon starvation conditions. We have added this discussion to the revised manuscript.

11. Fig 4A and D, authors might want to double check the significance level of the observed differences. Additionally, the chart in Fig4D contains only one series, while it is annotated as comparison of root lengths with and without NaCl.

>We verified that the results presented in Figure 4 are statistically significant. To improve the presentation of the data for root length assay, we have now used violin plots with all data points for each genotype and a box plot to show the data (Figure 4d, Supplementary Figure 4h). The presented data displays the ratio of root length with and without NaCl (the individual data points with NaCl were divided by the averaged root length without NaCl). The comparison of absolute root length is shown in Supplementary Figure 4h.

The assay was repeated reproducibly and the raw data for the one representative result shown in Figure 4 and Supplementary Figure 4 can be found in the "Source Data File". In addition, we have included the analysis of the second CRISPR line (*calb1-2*) in Supplementary Figure 4i and j, which confirms that *calb1* mutant lines are significantly more sensitive to the salt treatment when compared with the wild type.

12. Figure 4 should probably also contain side by side comparison of CaLB and ATG knockout seedlings sensitivity to NaCl?

>We thank the reviewer for this suggestion and have now included a side-by-side measurement of an *atg10* mutant (*atg10-1*) together with the second CRISPR line *calb1-2* and the wild type in Supplementary Figure 4i and j. Both *atg10-1* and *calb1-2* show increased salt sensitivity when compared to the wild type.

13. It would be most instructive to see if introducing *calb* mutation into ATG-deficient background provides no additional phenotypes. However, establishing such lines would cause an unreasonable delay for the publication and is not recommended if the required material is not available.

>We agree with the reviewer that testing the genetic interaction between *calb1* and *atg* mutants would be very interesting. In the limited time, however, it was not possible to generate a double homozygous mutant and thus the analysis will have to await future studies.

14. The scale bar size is missing for the Fig 5A-C

>Thank you for pointing this out. The size of the scale bar is now included.

15. For the Figure 5 it would be instructive to indicate what was counted as puncta and why (maybe an illustrative example could be shown in a magnified inset with arrowheads pointing at puncta).

>For counting the foci, thresholding was applied to the whole image cutting off low intensity signals. The size and circularity filtering applied for each experiment is described in detail in the Method section "Microscopy and image analyses".

16. Fig 5d might have a mistake in y axis labelling. Judging by Figs5a-c, it is unlikely the authors observed 40 autophagosomes/10 square um or is it 10 000 um²?

s

>The comma (,) in "10,000 μm²" was used as a thousands separator and indicates "10000 μm²". We are aware that the use of comma and period can be reversed according to the languages used. Though we believe that the use of comma as a thousands separator is appropriate when describing the numbers in English, we now removed the comma and instead inserted a space as a separator to avoid any confusion.

17. Fig 5e is not sufficiently described in the text to understand authors opinion: why does expression of CaLB go down after 90 minutes? How is this related to CaLB role in autophagosome biogenesis? In the FigS6b the ATG8e expression pattern seems to closely resemble that of CaLB.

>We have now described the results more in detail and included a immunoblot to analyzing the protein amount of GFP-CaLB₁, ALIX-GFP, and GFP-ATG8 upon salt treatment in Figure 4b. Though among the three, only *CaLB1* expression is salt induced, all protein levels are increased upon salt stress, suggesting that the increased autophagic activity may lead to stabilization of these proteins. Although we can only speculate, the transient increase in *CaLB1* protein level might be sufficient for the function of *CaLB1* in salt-induced autophagy.

We agree with the reviewer that the expression pattern of ATG8e resembles *CaLB1*, however, with the number of replicates used for the analysis, there was no significant difference to the untreated sample.

18. The way the co-localization of ALIX and CaLB to autophagosomes is presented does not align very well with the fractionation experiments shown on the same figure.

It would be informative to include co-localization of CaLB and ALIX under the same salt stress conditions.

The relevance of the localization phenotypes quantified in Fig 5j and n are not discussed, furthermore, ring-like localization of ALIX seems to contradict the suggested mechanism on the final scheme.

Possibly, the charts with colocalization phenotypes are better suited for supplementary figures/future perspective?

>We agree with the reviewer that the microsomal fractionation alone is not very informative. To more accurately analyze the association of ALIX and CaLB₁ to autophagosomes, we performed fractionation experiments and included the data in Figure 5g. The result shows that CaLB₁ is enriched in the fraction where autophagosome markers (ATG8 and NBR₁) are present, while UGPase (cytosol) does not show enrichment.

We now included the quantification results of colocalization between ALIX and CaLB₁ under salt stress condition in Figure 5e.

The percentage of GFP-ALIX showing ring-like structure is low (6.9 %) compared to those showing a dot on the ring localization. Therefore, we believe that the majority of ALIX localizes to distinct foci on the autophagosomal membrane as CaLB.

Following the reviewer's suggestions, the analysis of the colocalization dynamics is now moved to Supplementary Figure 5c, d and e.

19. Authors do not explain their choice of ATG8 orthologs for different experiments shown in Figure 5. For example, why is panel g showing ATG8a and panel k is showing ATG8i?

>We thank the reviewer for pointing this out. In principle, we have started the experiments with a published ATG8a line (Reyes et al., 2011, The Plant Cell), as later studies have shown that CaLB interacts with all three ATG8 clades, we have generated additional ATG8 lines. We have not detected any clade-specific co-localization of ATG8s with CaLB or ALIX. For colocalization studies, we now included data with ATG8i that was generated in this study and moved the data with ATG8a, if available, to the Supplementary Figures 5b and f. For biochemical analyses, we used GFP-ATG8a, as it is the homolog used in the majority of past studies.

20. Lines 223-227: it is a bit surprising that authors decided to use conditions that they previously show to cause no phenotype in the *calb* loss-of-function mutant. It is not clear why this experiment is mentioned together with the NaCl treatment. Do authors see significance in this expectedly negative result? If yes, could authors elaborate on it?

>To focus on salt-induced autophagy we have removed part of the experiments and rephrased the text. GFP cleavage assays and Proteinase K assays (Figures 8b, c, i and j) added in the revised manuscript show that there is a maturation/delivery/degradation delay of GFP-ATG8 in the *calb1* mutant when compared with the wild type.

21. To aid interpretation of the data, panels c and d in the Figure 6 require presentation of individual channels in addition to the merged image. It would be also great to have zoomed-in scans of autophagosomes, similar to those shown in Fig 5j. Furthermore, panel illustrates side by side

micrographs made on cortical cytoplasm, containing a lot of puncta and micrograph made on the optical section including large empty vacuole. The latter would naturally contain less puncta and thus quantification per total area might be skewed.

This figure should also include similar assessment performed using ALIX marker line.

>We thank the reviewer for this comment. We have now included the individual channels for Figure 6. We now selected a micrograph from comparable regions in both conditions. The quantification was performed on similar regions in at least 4 individual seedlings.

For GFP-ALIX, we conducted a similar experiment and have included the data in the revised manuscript as Figure 6d and f. Similar to that of CaLB₁-GFP, the number of GFP-ALIX-positive foci became significantly smaller upon 1,6-Hexanediol treatment, whereas mRFP-ATG8i foci remained. These results show that CaLB and ALIX foci are at least partially phase-separated molecular condensates.

22. Lines 276-277: MDC has been demonstrated to be a questionable method for autophagy monitoring (<https://doi.org/10.1093/pcp/pcuo41>, doi: 10.1080/15548627.2020.1797280.) It has to be performed with appropriate controls, including staining of ATG knockouts as a negative control and showing co-localization of GFP-ATG8 and MDC in the WT background under used conditions. Additionally, measuring area of a 3D cluster of potentially autophagic bodies be not representative of autophagic activity. Authors might want to double check statistical significance of values show in Fig. 8b.

>We thank the reviewer for this comment and agree that additional controls should be included. Following the reviewer's suggestion, we have now included two control experiments in Figure 8f and Supplementary Figure 8a and b: MDC staining in an *atg10-1* mutant after NaCl treatment and the colocalization of GFP-ATG8 with MDC in the wild type background upon NaCl treatment.

Reviewer #3 (Remarks to the Author):

Mosesso et al. identified a new factor that interacts with ALIX via yeast two-hybrid screen and confirmed their interaction. They demonstrated that as the name suggests, CaLB binds to lipids in a calcium-dependent manner.

They also found that CaLB interacts with ATG8 and localizes on salt-induced autophagosomes. The authors further showed that CaLB co-phase separates with ALIX in vitro and forms condensates in vivo. Finally, they showed that CaLB is required for the recruitment of ESCRT-III to autophagosomes. The findings of this work provide insights into how ESCRT is recruited to autophagosome to help maturation, in particular, under stress conditions. The CaLB-ALIX module could be one of the routes that relay stress signal to protein quality control via autophagy. However, based on current data, I am concerned about the significance of CaLB-ALIX module for the following reasons. As the author also mentioned in the discussion, others pathways such as FREE1-mediated recruitment of ESCRT-III to autophagosome also exist.

>We thank the reviewer for the suggestions that helped us improve our manuscript. We reduced part of the data regarding the Ca²⁺-dependent lipid binding and performed additional experiments that implicate CaLB1 in autophagosome maturation. The point-by-point answer to the points raised by the reviewer can be found below:

1. The importance of CaLB in salt stress tolerance is questionable. The authors stated that mutant seedlings showed a significant reduction of primary root length compared with the wild type but the photo shown in Figure 4c seems not supporting this conclusion. I did a rough measurement, the resulting normalized ratio is similar in wild type and mutant. In any case, the current does not support the necessity of CaLB for salt stress tolerance.

>To improve the presentation of the data, we have now used violin plots with all data points and a box plot to show the data (Figure 4d, Supplemental Figure 4h, i and j). The representative photograph with several seedlings is only part of the analysis. The assay was performed three times reproducibly and the measurement of one data set with at least 50 seedlings for each genotype is shown. The raw data is presented in the "Source Data File". In addition, we have now included the analysis of the second CRISPR line together with an *atg10* mutant in Supplementary Figure 4i and j, that shows that both the second *calb1* mutant *calb1-2* as well as *atg10* are significantly more sensitive to the salt treatment when compared with the wild type. We believe that the improved data presentation, the provided raw data, and the analysis of the second *calb1* mutant line all indicates that CaLB1 is implicated in salt stress tolerance.

2. The role of CaLB in the efficient maturation of autophagosomes is rather speculative. Electron microscopic experiments should be performed to directly observe whether autophagosome are immature in *calb* mutant background.

>We thank the reviewer for this comment and the suggestions. Our electron microscopic analysis turned out to be technically difficult to collect enough number of autophagosomes to analyze the autophagosomal structure and electron tomography was because of the time-intensive nature not

optimal for this purpose for us. We thus performed a biochemical analysis to assess the maturation status of autophagosomes as published before in yeast (Zhou et al, 2019, JCB) and successfully applied in plant cells in a recent publication (Zhao et al. 2022, JCB).

We isolated autophagosomes from wild type and *calb1* mutant lines and analyzed the protection of GFP-ATG8a upon Proteinase K treatment. The new data is now included in Figure 8i and shows that more GFP-ATG8a is digested by Proteinase K, suggesting that there is more exposed GFP-ATG8a in *calb1* mutant seedlings. Together with the localization of CaLB1 on autophagosomes, its interaction with ALIX, and the *calb1* mutant phenotype, we believe the data supports that CaLB1 is implicated in the maturation of autophagosomes.

3. As the title suggests that phase separation of CaLB modulates autophagosome maturation, but there is no evidence support this. CaLB may phase separate in vivo, but genetic evidence disrupting phase separation is required to prove whether phase separation is necessary for its function in autophagosome maturation.

>We agree with the reviewer and performed the following additional experiments and included the data in the revised manuscript:

a. We performed qRT-PCR on all ATG8s upon salt treatment and showed that there is no apparent difference between the wild type and the *calb* mutant (Supplemental Figure 8c).

b. We quantified the number of GFP-ATG8a-positive foci in wild type and *calb1* (Figure 8d and e). Upon salt treatment, *calb1* has a higher induction of GFP-ATG8 foci (ratio with/without NaCl) after salt stress.

c. We isolated membrane fractions enriched in autophagosomes from wild type and *calb1* mutant seedlings and performed a proteinase K protection assay (Figure 8i). The results show that there are more GFP-ATG8 is less protected in the *calb1* mutant when compared with the wild type, suggesting that *calb1* has less matured autophagosomes than the wild type.

c. We monitored the cleavage of GFP from GFP-ATG8a in wild type and *calb1* mutant seedlings upon salt treatment (Figure 8j). There is less release of free GFP in the *calb1* mutant compared with the wild type, suggesting that there is less efficient transport/degradation of GFP-ATG8 containing autophagosomes in the *calb1* mutant seedlings.

4. The authors used suspension cells in which CaLB is transiently knocked down, the efficiency of which is low, to test the colocalization of ESCRT with autophagosomes. As we know that the concentration is key for phase separation, but all proteins were overexpressed. The conclusions drawn from these data are not convincing. These experiments should be done in CRISPR mutant background.

>We thank the reviewer for this suggestion. Formally published VPS2.1 constructs were not functional and therefore we have generated a new *VPS2.1pro: VPS2.1-mRFP* construct with a long linker and

verified its functionality by complementation of an embryo lethal *vps2.1* mutant (Supplemental Figure 8 f to h).

We then analyzed the co-localization of VPS2.1-mRFP with GFP-ATG8a in the wild type and in the *calb1* mutant. Colocalization of VPS2.1-mRFP with GFP-ATG8a could be observed in both genotypes, however, the number of colocalization incidents was too low for a meaningful comparison.

5. In line with my concern, I find it difficult to understand that knock down of CaLB increased the number of autophagosomes per cell and more autophagosomes colocalized with ESCRT-III, but the authors think phase separation of ALIX and CaLB on the autophagosomal membrane could facilitate ESCRT-III recruitment. It is also overstated to say CaLB phase separation could facilitate membrane scission.

>As the number of autophagosomes per protoplast were very low in the experiment, we removed the data from the revised manuscript. As described in the answer to point 3, we have added new data supporting the function of CaLB1 in autophagy and rephrased the text.

6. The increased number of CaLB-GFP and GFP-ALIX foci correlated with induced CaLB transcript level by salt, but it is not clear whether CaLB expression is the cause. As it is a calcium-dependent lipid binding, to test whether the increase of intracellular foci positive for CaLB-GFP, GFP-ATG8a, and GFP-ALIX under salt stress is due to calcium, the authors can use calcium channel inhibitors such as LaCl₃.

>We followed the advice of the reviewer and performed experiments to test whether the calcium-dependent lipid binding of CaLB1 is induced using Lanthanum(III) chloride (LaCl₃) and included the results in Supplementary Figure 4k and l. The analysis shows that treatment with LaCl₃ does not affect the number of CaLB1 foci.

7. How does the authors define CaLB-GFP signals that are proximal to mRFP-ATG8 signal shown in Figure 5g, they might be just close to each other, not necessarily associating, as from the biophysical perspective, the association of a condensate with similar size to the autophagosome to the autophagosomal membrane is unfavorable.

The data showed that 15.0%, 8.3%, and 15.5% of structures positive for ATG8a, ATG8e and ATG8i, respectively, colocalized with CaLB-GFP, which is not good colocalization, leading to the conclusion that CaLB localizes to autophagosome, but when 16.6% of CaLB-GFP foci co-localized with ARA6-mRFP in the presence of Wortmannin, the authors concluded that CaLB does not localize on endosomes with high frequency.

Higher resolution is required to show 35.3% of CaLB-mRFP foci colocalize with GFP-ALIX.

>We thank the reviewer for this comment and now included a higher resolution image of GFP-ALIX/CaLB1-mRFP. We also rephrased the sentence regarding the colocalization of CaLB1-GFP on WM-induced structures.

8. Why the authors used BFA and WM when observing colocalization of CaLB with SYP43 and ARA, respectively? Intuitively, the observation of native endosomes does not need these drugs, therefore, the conclusion that that CaLB does not localize on endosomes with high frequency is not confident.

>Proteins localizing on endosomal membranes will relocate to BFA bodies and WM rings upon treatment with BFA and WM, respectively. The experiments shown in Supplementary Figure 3 a and b were performed to analyze whether these drugs have an impact on CaLB₁ localization, which would indicate that CaLB₁ are associated with endosomes. Although we saw an increase in CaLB₁ localization to WM rings, the BFA treatment did not affect the localization of CaLB₁.

We thus conclude that CaLB₁ is not stably and abundantly associated with endosomes. To avoid any misunderstandings, we rephrased the text (lines 172-176).

9. The quality of immuno-electron microscopy on CaLB-GFP is too poor. In theory, it is possible to see a high density region around the gold particle signal if CaLB does form condensates on autophagosome membrane.

>Following the advice of the reviewer, we reanalyzed the immune-TEM images and also imaged further sections. We have now replaced the immune-electron micrograph in Supplementary Figure 5g with an image on which the gold particles are better visible. In our preparations, we could not detect high density regions around the gold particle signals, which might be due to the fixation and processing required for immune electron microscopy.

10. When observing ALIX localization on autophagosomes, the authors were able to see the ring-like structures, but when observing CaLB, ATG8 signal is always a filled structure, is it a matter of resolution or other reasons. Also, better quality images are needed to convince me that ALIX also form puncta similarly to CaLB on autophagosome.

>We thank the reviewer for pointing this out. The original images of CaLB₁/ATG8 co-expressing line were performed using the confocal microscopy without AiryScan whereas for ALIX/ATG8 AiryScan was used due to the low fluorescence intensity of ALIX-GFP. We now retook the images using the AiryScan modus of the microscope and present comparable images of CaLB₁/ATG8 and ALIX/ATG8. The new images are presented in Figures 5f and j.

11. When interrogating the property of CaLB condensates, 5% 1,6-hexanediol is already very strong, it is not enough to judge if they are liquid-like or not only based on the dissolution. A further experiment could be wash out of the hexanediol to see whether CaLB condensates reform in the same root. FRAP experiment should also be performed.

>We thank the reviewer for this comment. In the literature, 1,6-hexanediol is used typically at a concentration between 5 to 10%. We are aware that a prolonged (1h-) treatment with hexanediol can cause side effects (Düster et al., 2021, JBC) and therefore imaged the seedling roots immediately after

application with 1,6-hexanediol. We have now also included an experiment with GFP-ALIX in Figures 6d and f. Our control experiments show that the less polar 1,2,6-hexanetriol has a milder effect on the CaLB1 foci and that the number of ATG8-positive foci does not change upon treatment with 1,6-hexanediol. These results thus suggest that CaLB1 and ALIX foci have a different nature than the ATG8 foci and that they most likely represent condensates that can be dissolved with 1,6-hexanediol.

According to the reviewer's suggestion, we have performed a hexanediol washout analysis and the new data is included in Supplementary Figure 6a. The results shows that CaLB1 foci reforms in the cytosol after washout of 1,6-hexanediol.

We tried to perform FRAP experiments on CaLB1 and ALIX foci *in vivo*. However, due to the dynamic nature of the foci, weak signals, and the size, it was not possible to collect enough data points to confidently show the FRAP efficiency.

12. The *in vitro* data support that CaLB enhances the phase separation of ALIX, the authors should also test whether the condensation of ALIX *in vivo* depends on CaLB. This should be straightforward to observe in the CRISPR mutant background.

>As described in the answer to Reviewer 1 point 10, despite multiple attempts, we could not combine the published GFP-ALIX line (Cardona-Lopez et al, 2015, The Plant Cell) with the *calb1* CRISPR mutants generated in this study, most probably due to genetic linkage.

We used the published GFP-ALIX plasmid for transformation, however, could not obtain a line that expressing GFP-ALIX.

We therefore performed additional experiments to investigate the ALIX foci formation and the influence of CaLB1 *in vivo*. In Arabidopsis root-derived cell culture, we expressed GFP-ALIX together with a CRISPR construct that targets CaLB1. As a control a mutated CRISPR construct was used. We have analyzed over 100 cells in each combination and the quantification of the images shows that in the presence of the CaLB1 CRISPR construct targeting CaLB1, the number of ALIX foci per cell is significantly reduced (Figure 6i and j). In the same set up, we tried to co-express ALIX with ATG8, however, due to the low number of autophagosomes per cell, a reliable quantification was not possible.

13. For the FRAP experiment *in vitro*, selected regions were bleached with 100 iterations at a laser power of 100%, I think this could be over-bleached. Sometimes, overbleach leads to less dynamic and recovery.

>We thank the reviewer for this comment. We used a FRAP condition that has been optimized for labelled proteins used in *in vitro* condensate assays with our confocal set up. We agree with the reviewer that while the recovery of CaLB1 is dynamic with high recovery, the recovery of ALIX is less efficient. However, we believe this to reflect the nature of the ALIX protein behavior in condensates, rather than this being the result of overbleaching.

In a recent publication using human ALIX (Elias et al., 2023, Science Advance), it was reported that the recovery of human ALIX was at a “neglectable” level, and while ALIX forms probably transiently phase separate into liquid droplets, they probably readily reach a more rigid state, that leads to subsequent formation of ALIX filaments. The formation of ALIX filaments is considered to be favorable for the recruitment of the ESCRT-III machinery, as it increases the contact surface with the ESCRT-III which also has a filamentous architecture.

REVIEWER COMMENTS

Reviewer #1 (Remarks to the Author):

The authors have address most of my concerns, however, I still have some further comments related to their data interpretation and some new data added in the revision.

1. The authors mentioned that these proteins affect phagophore closure, but to me is statement is not accurate enough without convincing TEM analysis. The results only show autophagosome formation is inhibited, which can happen at any stage. I would suggest the authors to tune it down and rephrase it in the abstract and main text.

2. Figure 1h, the co-localization image is not very clear, please show each channel individually, and provide zoom in images if possible.

3. Figure S5 should be part of the main figure, the results are very informative for readers to understand the manuscript.

4. Figure 6i-j, why not use protoplasts from the calb1 mutant which is available? Please explain. Transient expression a Crispr construct in protoplasts is not a very reliable approach in my opinion, judging from the date distribution in Figure 6j, the variation is very significant.

Reviewer #2 (Remarks to the Author):

Authors have addressed all our concerns. There are just a few minor comments:

1. Neither the title nor abstract refer to salt-stress induced autophagy. Although one could consider salt-stress to be a more narrow topic compared to autophagy in general, the autophagy mode-dependent roles of CaLB that authors report might still be a more exciting discovery and an

interesting topic for the discussion chapter. That being said, we leave it to the authors to decide whether such presentation of their results would be most informative.

2. Authors provided more detailed background information on CaLB, which is very helpful. It could be also of help to introduce the BRO1 domain for ALIX and its' distinction from the yeast ortholog, as well as the V domain.

3. Related to the comment 2: we could not locate the information about Y2H being performed on plant orthologs only. The annotations on the panels b and c of the Figure 1 look much more informative, however P and PRD seem to be used interchangeably for CaLB1, while V domain is not introduced for ALIX and BRO1 only mentioned as the deleted part, without explanation of the hypothesis behind such deletion. A more detailed information on these domains and consistent labeling could greatly aid interpretation of the ALIX/CaLB interaction assay results.

4. Maybe authors would consider it informative to provide a more detailed reasoning behind selecting putative regions of CaLB1 that are responsible for interaction with ATG8. At the moment the logic behind the mutagenesis is somewhat obscure. Additionally, it would be interesting to know authors opinion on how the identified aa of CaLB warrants interaction with ATG8 while not being a part of canonical or known non-canonical types of AIM. For the same AlphaFold scheme: would it be possible to also label the aa numbers and select a highlighting color different from grey.

5. Figure 5 legend seems to list more panels than are present on the figure.

6. Methods chapter might need some proof reading. For example, chapter “Immunoblotting, protein staining, and ultracentrifugation” seems to not contain any information on ultracentrifugation.

Reviewer #3 (Remarks to the Author):

The authors performed additional experiments and explanations that addressed most of my concerns except following two concerns:

Comment to the response to my major point 2:

The authors performed biochemical characterization of autophagosomes from wild type and calb1 mutant lines to assess the maturation status of autophagosomes. This is very helpful. However, since this experiment is very tricky, the authors should present this result very carefully. The wild type and the mutant should be loaded in parallel on the same gel. Without any treatment, ATG8 protein level should be comparable between wild type and calb1 mutant (I did not find the data that calb1 mutation alters ATG8 protein accumulation). With three biological replicates, audiences will be more convinced with the role of CaLB1 in autophagosome maturation. This is not trivial because it is a key point of this manuscript.

Comment to the response to my major point 3:

My concern was whether the phase separation of CaLB1 is important. The additional experiments did help to prove the CaLB1 protein per se is important, but does it really need to phase separate in order to fulfill its function? Otherwise, this story does not benefit from describing phase separation.

REVIEWER COMMENTS

We thank all the reviewers for reading our revised manuscript and are grateful for their constructive suggestions. Below is our point-by-point response to the points raised by the reviewers.

Reviewer #1 (Remarks to the Author):

1. The authors mentioned that these proteins affect phagophore closure, but to me is statement is not accurate enough without convincing TEM analysis. The results only show autophagosome formation is inhibited, which can happen at any stage. I would suggest the authors to tune it down and rephrase it in the abstract and main text.

>We had analyzed TEM data, however, have not included them in the previous versions of the manuscript. The ultrathin sections (50 nm) for TEM covers only a small region of the relatively large salt-induced autophagosome (~1 μm in diameter), and seemingly “closed” phagophore membranes still can have an opening outside of the analyzed section. That being said, we agree that if there are more “open” autophagosomes, the chance to find the opening in ultra-thin sections will also increase. We have thus now included the TEM analysis carried out on ultrathin sections of roots from wild-type and *calb1-1* seedlings in Figure 8i and Supplementary Figure 8h. The data show that there are more unclosed autophagosomes in *calb1-1* compared with the wild type, further supporting that CaLB1 is required for the maturation of autophagosomes.

2. Figure 1h, the co-localization image is not very clear, please show each channel individually, and provide zoom in images if possible.

>Figure 1h has been changed according to the previous suggestion of the reviewer and individual channels as well as a magnification of the colocalizing puncta is shown.

3. Figure S5 should be part of the main figure, the results are very informative for readers to understand the manuscript.

>Figure 5 was modified by adding panel c, d, e, g and h from Supplementary Figure 5. Figure 5 and Supplementary Figure 5 legends were modified accordingly.

4. Figure 6i-j, why not use protoplasts from the *calb1* mutant which is available? Please explain. Transient expression a Crispr construct in protoplasts is not a very reliable approach in my opinion, judging from the date distribution in Figure 6j, the variation is very significant.

>We thank the reviewer for the suggestion. We have now included an experiment using protoplasts from wild-type and *calb1-1* mutant roots. This experiment, which is now included as Supplementary Figure 6c-e. The results show the same tendency as in Figure 6j. The variation of the number of ALIX foci probably originates from the fact that ALIX is not only active on autophagosomes but also on endosomal membranes. As CaLB does not localize on endosomes, most probably the absence of CaLB function does not affect the endosomal localization and function of ALIX.

Reviewer #2 (Remarks to the Author):

1. Neither the title nor abstract refer to salt-stress induced autophagy. Although one could consider salt-stress to be a more narrow topic compared to autophagy in general, the autophagy mode-dependent roles of CaLB that authors report might still be a more exciting discovery and an interesting topic for the discussion chapter. That being said, we leave it to the authors to decide whether such presentation of their results would be most informative.

>We agree with the reviewer that the autophagy mode-dependent role of CaLB1 is an interesting aspect for our manuscript. We, now refer to salt-induced autophagy in the abstract (line 39 and lines 40-41) as well as in the introduction (lines 47-51 and line 95-96) and we addressed the possible mode-dependent role of CaLB1 and ALIX in the discussion (lines 492-495 and 537-539).

2. Authors provided more detailed background information on CaLB, which is very helpful. It could be also of help to introduce the BRO1 domain for ALIX and its' distinction from the yeast ortholog, as well as the V domain.

>We introduced *Arabidopsis* ALIX in the introduction section describing the similarity with its mammalian and yeast homologs. Moreover, we described the role of the different domains of ALIX (lines 72-79).

3. Related to the comment 2: we could not locate the information about Y2H being performed on plant orthologs only. The annotations on the panels b and c of the Figure 1 look much more informative, however P and PRD seem to be used interchangeably for CaLB1, while V domain is not introduced for ALIX and BRO1 only mentioned as the deleted part, without explanation of the hypothesis behind such deletion. A more detailed information on these domains and consistent labeling could greatly aid interpretation of the ALIX/CaLB interaction assay results.

>We included now more detailed information about ALIX domains in the introduction section (lines 72-79) and we provided a more consistent labeling on panel b and c for Figure 1. Additionally, we indicate that *Arabidopsis* ALIX was used for the Y2H assay (lines 147-152).

4. Maybe authors would consider it informative to provide a more detailed reasoning behind selecting putative regions of CaLB1 that are responsible for interaction with ATG8. At the moment the logic behind the mutagenesis is somewhat obscure. Additionally, it would be interesting to know authors opinion on how the identified aa of CaLB warrants interaction with ATG8 while not being a part of canonical or known non-canonical types of AIM. For the same AlphaFold scheme: would it be possible to also label the aa numbers and select a highlighting color different from grey.

> We thank the reviewer for the insights on CaLB1-ATG8 interaction. We explain now in the text how the four putative AIMs were selected (lines 197-199). We also modified the Supplementary Figure 3e and the respective part in the figure legend by including the amino acid numbers and changing the grey color to make the region in the PRD domain of CaLB1 more visible, as suggested by the reviewer.

5. Figure 5 legend seems to list more panels than are present on the figure.

>Figure 5 was modified by adding panel c, d, e, g and h from Supplementary Figure 5. Figure 5 and Supplementary Figure 5 legends were modified accordingly.

6. Methods chapter might need some proof reading. For example, chapter “Immunoblotting, protein staining, and ultracentrifugation” seems to not contain any information on ultracentrifugation.

> We corrected the main text in the Methods chapter.

Reviewer #3 (Remarks to the Author):

Comment to the response to my major point 2: The authors performed biochemical characterization of autophagosomes from wild type and calb1 mutant lines to assess the maturation status of autophagosomes. This is very helpful. However, since this experiment is very tricky, the authors should present this result very carefully. The wild type and the mutant should be loaded in parallel on the same gel. Without any treatment, ATG8 protein level should be comparable between wild type and calb1 mutant (I did not find the data that calb1 mutation alters ATG8 protein accumulation). With three biological replicates, audiences will be more convinced with the role of CaLB1 in autophagosome maturation. This is not trivial because it is a key point of this manuscript.

>We have now averaged the data of four experiments and all data points are shown.

Comment to the response to my major point 3: My concern was whether the phase separation of CaLB1 is important. The additional experiments did help to prove the CaLB1 protein per se is important, but does it really need to phase separate in order to fulfill its function? Otherwise, this story does not benefit from describing phase separation.

>We thank the review of this insight and rephrased part of the conclusion. Our in vitro experiments show that CaLB1 positively impacts phase separation of ALIX. Protoplast experiments from Arabidopsis cell culture and wild-type as well as *calb1-1* mutant seedlings show that the presence of CaLB1 is important for the formation ALIX condensates, and we could also show that CaLB1 and ALIX localize on condensates in vivo. Our analyses have also shown that loss of CaLB1 leads to less efficient maturation in autophagosomes. Based on these results, we propose a model that CaLB1 and ALIX assemble together into condensates localized on autophagosomal membrane and that they are important for the ESCRT-mediated autophagosomal maturation.

EVIEWERS' COMMENTS

Reviewer #1 (Remarks to the Author):

The authors have addressed all my concerns, this is an interesting piece of work, congratulations!

Reviewer #2 (Remarks to the Author):

Authors addressed all our questions and comments in full and introduced significant amount of additional data corroborating their original conclusions.

There are some minor technical corrections required. For example, descriptions for panels FigS8c and d seem to be mixed up in the corresponding figure legend

Reviewer #3 (Remarks to the Author):

The authors have adequately addressed my concerns. The manuscript can be considered for publication.

REVIEWER COMMENTS

We would like to thank all the reviewers for their comments and suggestions throughout the review process, which helped us to improve our manuscript considerably.

Reviewer #1 (Remarks to the Author):

The authors have addressed all my concerns, this is an interesting piece of work, congratulations!

Reviewer #2 (Remarks to the Author):

Authors addressed all our questions and comments in full and introduced significant amount of additional data corroborating their original conclusions.

There are some minor technical corrections required. For example, descriptions for panels FigS8c and d seem to be mixed up in the corresponding figure legend.

>The figure caption for Sup Fig. 8c and d are now corrected.

Reviewer #3 (Remarks to the Author):

The authors have adequately addressed my concerns. The manuscript can be considered for publication.